# A Heat Diffusion Perspective on Geodesic Preserving Dimensionality Reduction

**Guillaume Huguet**[1][*]    **Alexander Tong**[1][*]    **Edward De Brouwer**[2][*]

**Yanlei Zhang**[1]    **Guy Wolf**[1]    **Ian Adelstein**[2][†]    **Smita Krishnaswamy**[2][†]
[1]Université de Montréal; Mila - Quebec AI Institute    [2] Yale University

## Abstract

Diffusion-based manifold learning methods have proven useful in representation learning and dimensionality reduction of modern high dimensional, high throughput, noisy datasets. Such datasets are especially present in fields like biology and physics. While it is thought that these methods preserve underlying manifold structure of data by learning a proxy for geodesic distances, no specific theoretical links have been established. Here, we establish such a link via results in Riemannian geometry explicitly connecting heat diffusion to manifold distances. In this process, we also formulate a more general heat kernel based manifold embedding method that we call *heat geodesic embeddings*. This novel perspective makes clearer the choices available in manifold learning and denoising. Results show that our method outperforms existing state-of-the-art in preserving ground truth manifold distances, and preserving cluster structure in toy datasets. We also showcase our method on single cell RNA-sequencing datasets with both continuum and cluster structure, where our method enables interpolation of withheld timepoints of data. Finally, we show that parameters of our more general method can be configured to give results similar to PHATE (a state-of-the-art diffusion based manifold learning method) as well as SNE (an attraction/repulsion neighborhood based method that forms the basis of t-SNE).

## 1   Introduction

The advent of high throughput and high dimensional data in various fields of science have made dimensionality reduction and visualization techniques an indispensable part of exploratory analysis. Diffusion-based manifold learning methods, based on the data diffusion operator, first defined in [5], have proven especially useful due to their ability to handle noise and density variations while preserving structure. As a result, diffusion-based dimensionality reduction methods, such as PHATE [22], T-PHATE [3], and diffusion maps [5], have emerged as methods for analyzing high throughput noisy data in various situations. While these methods are surmised to learn manifold geodesic distances, no specific theoretical links have been established. Here, we establish such a link by using Varadhan's formula [34] and a parabolic Harnack inequality [17, 25], which relate manifold distances to heat diffusion directly. This lens gives new insight into existing dimensionality reduction methods, including when they preserve geodesics, and suggests a new method for dimensionality reduction to explicitly preserve geodesics, which we call *heat geodesic embeddings*[3]. Furthermore, based on our understanding of other methods [22, 5], we introduce theoretically justified parameter choices that

---

[*]Equal contribution

[†]Co-senior authors

[3]`https://github.com/KrishnaswamyLab/HeatGeo`

allow our method to have greater versatility in terms of distance denoising and emphasis on local versus global distances.

Generally, data diffusion operators are created by first computing distances between datapoints, transforming these distances into affinities by pointwise application of a kernel function (like a Gaussian kernel), and then row normalizing with or without first applying degree normalization into a Markovian diffusion operator $\boldsymbol{P}$ [5, 9, 14, 21, 33]. The entries of $\boldsymbol{P}(x, y)$ then contain probabilities of diffusing (or random walk probabilities) from one datapoint to another. Diffusion maps and PHATE use divergences between these diffusion or random walk-based probability distributions $\boldsymbol{P}(x, \cdot)$ and $\boldsymbol{P}(y, \cdot)$ to design a diffusion-based distance that may not directly relate to manifold distance. Our framework directly utilizes a heat kernel based distance, and offers a more comprehensive perspective to study these diffusion methods. By configuring parameters in our framework, we show how we can navigate a continuum of embeddings methods from PHATE [22] to Stochastic Neighbor Embedding (SNE) [11].

In summary, our contributions are as follows:

- We define the *heat-geodesic* dissimilarity based on Varadhan's formula and the two-sided heat kernel bounds.

- Based on this dissimilarity, we present a versatile geodesic-preserving method for dimensionality reduction which we call *heat geodesic embedding.*

- We establish a relationship between diffusion-based distances and the heat-geodesic dissimilarity.

- We establish connections between our method and popular dimensionality reduction techniques such as PHATE and SNE, shedding light on their geodesic preservation and denoising properties based on modifications of the computed dissimilarity and distance preservation losses.

- We empirically demonstrate the advantages of Heat Geodesic Embedding in preserving manifold geodesic distances in several experiments showcasing more faithful manifold distances in the embedding space, as well as our ability to interpolate data within the manifold.

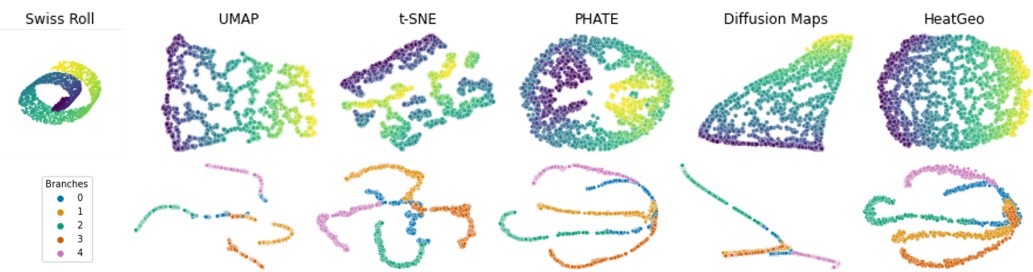

Figure 1: Embeddings of the Swiss roll (top) and Tree (bottom) datasets for different manifold learning methods. Our HeatGeo method correctly unrolls the Swiss roll while t-SNE and UMAP create undesirable artificial clusters.

## 2 Preliminaries

First, we introduce fundamental notions that form the basis of our manifold learning methods: Varadhan's formula [34] on a manifold, diffusion processes on graphs, efficient heat kernel approximations, and multidimensional scaling [4, 12, 16].

**Varadhan's formula** Varadhan's formula is a powerful tool in differential geometry that establishes a connection between the heat kernel and the shortest path (geodesic) distance on a Riemannian manifold. Its versatility has led to widespread applications in machine learning [6, 10, 15, 27–29]. Let $(M, g)$ be a closed Riemannian manifold, and $\Delta$ the Laplace-Beltrami operator on $M$. The heat kernel $h_t(x, y)$ on $M$ is the minimal positive fundamental solution of the heat equation $\frac{\partial u}{\partial t} = \Delta u$ with initial condition $h_0(x, y) = \delta_x(y)$. In a $d$-dimensional Euclidean space the heat kernel is $h_t(x, y) = (4\pi t)^{-d/2} e^{-d(x,y)^2/4t}$ so that $-4t \log h_t(x, y) = 2dt \log(4\pi t) + d^2(x, y)$ and

we observe the following limiting behavior:

$$\lim_{t \to 0} -4t \log h_t(x, y) = d^2(x, y). \tag{1}$$

Varadhan [34] (see also [20]) proved that eq. 1 (now Varadhan's formula) holds more generally on complete Riemannian manifolds $M$, where $d(x, y)$ is the geodesic distance on $M$, and the convergence is uniform over compact subsets of $M$. A related result for complete Riemannian manifolds that satisfy the parabolic Harnack inequality (which includes convex domains in Euclidean space and Riemannian manifolds with non-negative Ricci curvature) is the two-sided heat kernel bound [25, 17], showing that for any $\epsilon \in (0, 1)$ there exist constants $c(\epsilon)$ and $C(\epsilon)$ such that

$$\frac{c(\epsilon)}{V(x, \sqrt{t})} \exp\left(-\frac{d(x, y)^2}{4(1 + \epsilon)t}\right) \le h_t(x, y) \le \frac{C(\epsilon)}{V(x, \sqrt{t})} \exp\left(-\frac{d(x, y)^2}{4(1 - \epsilon)t}\right), \tag{2}$$

where $V(x, \sqrt{t})$ is the volume of a ball or radius $\sqrt{t}$ centered at x. We denote this relation by $h_t(x, y) \simeq V(x, \sqrt{t})^{-1} \exp(-d(x, y)^2/t)$ and note that it again recovers eq. 1 in the $t \to 0$ limit, which is unsurprising as Varadhan's result holds more generally. More important for our purposes is that $h_t(x, y) \simeq V(x, \sqrt{t})^{-1} \exp(-d(x, y)^2/t)$ holds for $t > 0$ which will allow us to approximate geodesic distances $d(x, y)$ from a diffusion based estimation of the heat kernel $h_t(x, y)$ and the volume $V(x, \sqrt{t})$. In appendix C.3, we provide examples using inequality (2).

**Graph construction and diffusion**   Our construction starts by creating a graph from a point cloud dataset $\boldsymbol{X}$ of size $n$. We use a kernel function $\kappa : \mathbb{R}^d \times \mathbb{R}^d \to \mathbb{R}^+$, such that the (weighted) adjacency matrix is $\boldsymbol{W}_{ij} := \kappa(x_i, x_j)$ for all $x_i, x_j \in \boldsymbol{X}$. The kernel function could be a Gaussian kernel, or constructed from a nearest neighbor graph. The resulting graph $\mathcal{G}$ is characterized by the set of nodes (an ordering of the observations), the adjacency matrix, and the set of edges, i.e. pairs of nodes with non-zero weights. The graph Laplacian is an operator acting on signals on $\mathcal{G}$ such that it mimics the negative of the Laplace operator. The combinatorial graph Laplacian matrix is defined as $\boldsymbol{L} := \boldsymbol{Q} - \boldsymbol{W}$ and its normalized version as $\boldsymbol{L} = \boldsymbol{I}_n - \boldsymbol{Q}^{-1/2}\boldsymbol{W}\boldsymbol{Q}^{-1/2}$, where $\boldsymbol{Q}$ is a diagonal degree matrix with $\boldsymbol{Q}_{ii} := \sum_j \boldsymbol{W}_{ij}$. The Laplacian is symmetric positive semi-definite, and has an eigen-decomposition $\boldsymbol{L} = \Psi \Lambda \Psi^T$. Throughout the presentation, we assume that $\boldsymbol{Q}_{ii} > 0$ for all $i \in \{1, \dots, n\}$. The Laplacian allows us to define the heat equation on $\mathcal{G}$, with respect to an initial signal $\boldsymbol{f}_0 \in \mathbb{R}^n$ on $\mathcal{G}$:

$$\frac{\partial}{\partial t}\boldsymbol{f}(t) + \boldsymbol{L}\boldsymbol{f}(t) = \boldsymbol{0}, \ s.t. \quad \boldsymbol{f}(0) = \boldsymbol{f}_0 \quad t \in \mathbb{R}^+. \tag{3}$$

The solution of the above differential equation is obtained with the matrix exponential $\boldsymbol{f}(t) = e^{-t\boldsymbol{L}}\boldsymbol{f}_0$, and we define the heat kernel on the graph as $\boldsymbol{H}_t := e^{-t\boldsymbol{L}}$. By eigendecomposition, we have $\boldsymbol{H}_t = \Psi e^{-t\Lambda}\Psi^T$. The matrix $\boldsymbol{H}_t$ is a diffusion matrix that characterizes how a signal propagate through the graph according to the heat equations.

Other diffusion matrices on graphs have also been investigated in the literature. The transition matrix $\boldsymbol{P} := \boldsymbol{Q}^{-1}\boldsymbol{W}$ characterizing a random walk on the graph is another common diffusion matrix used for manifold learning such as PHATE [22] and diffusion maps [5]. It is a stochastic matrix that converges to a stationary distribution $\boldsymbol{\pi}_i := \boldsymbol{Q}_{ii}/\sum_i \boldsymbol{Q}_{ii}$, under mild assumptions.

**Fast computation of heat diffusion**   Exact computation of the (discrete) heat kernel $\boldsymbol{H}_t$ is computationally costly, requiring a full eigendecomposition in $O(n^3)$ time. Fortunately, multiple fast approximations have been proposed, including using orthogonal polynomials or the Euler backward methods. In this work, we use Chebyshev polynomials, as they have been shown to converge faster than other polynomials on this problem [13].

Chebyshev polynomials are defined by the recursive relation $\{T_k\}_{k \in \mathbb{N}}$ with $T_0(y) = 0$, $T_1(y) = y$ and $T_k(y) = 2yT_{k-1}(y) - T_{k-2}(y)$ for $k \ge 2$. Assuming that the largest eigenvalue is less than two (which holds for the normalized Laplacian), we approximate the heat kernel with the truncated polynomials of order $K$

$$\boldsymbol{H}_t \approx p_K(\boldsymbol{L}, t) := \frac{b_{t,0}}{2} + \sum_{k=1}^{K} b_{t,k}T_k(\boldsymbol{L} - \boldsymbol{I}_n), \tag{4}$$

where the $K + 1$ scalar coefficients $\{b_{t,i}\}$ depend on time and are evaluated with the Bessel function. Computing $p_K(\boldsymbol{L}, t)\boldsymbol{f}$ requires $K$ matrix-vector product and $K + 1$ Bessel function evaluation. The expensive part of the computation are the matrix-vector products, which can be efficient if the Laplacian matrix is sparse. Interestingly, we note that the evaluation of $T_k$ do not depend on the diffusion time. Thus, to compute multiple approximations of the heat kernel $\{p_K(\boldsymbol{L}, t)\}_{t \in \mathcal{T}}$, only necessitates reweighting the truncated polynomial $\{T_k\}_{k \in [1,...,K]}$ with the corresponding $|\mathcal{T}|$ sets of Bessel coefficients. The overall complexity is dominated by the truncated polynomial computation which takes $O(K(E + n))$ time where $E$ is the number of non-zero values in $\boldsymbol{L}$.

Another possible approximation is using the Euler backward method. It requires solving $K$ systems of linear equations $\boldsymbol{f}(t) = (\boldsymbol{I}_n + (t/K)\boldsymbol{L})^{-K}\boldsymbol{f}(0)$, which can be efficient for sparse matrices using the Cholesky decomposition [10, 28]. We quantify the differences between the heat kernel approximations in Appendix C.

**Metric multidimensional scaling** Given a dissimilarity function $d$ between data points, metric multidimensional scaling (MDS) [16] finds an embedding $\phi$ such that the difference between the given dissimilarity and the Euclidean distance in the embedded space is minimal across all data points. Formally, for a given function $d : \mathbb{R}^d \times \mathbb{R}^d \to \mathbb{R}^+$, MDS minimizes the following objective:

$$L(\boldsymbol{X}) = \left( \sum_{ij} \left( d(x_i, x_j) - \|\phi(x_i) - \phi(x_j)\|_2 \right)^2 \right)^{1/2}, \tag{5}$$

In metric MDS the solution is usually found by the SMACOF algorithm [30], or stochastic gradient descent [37].

## 3 Related Work

We review state-of-the-art embedding methods and contextualize them with respect to Heat Geodesic Embedding. A formal theoretical comparison of all methods is given in Section 5. Given a set of high-dimensional datapoints, the objective of embedding methods is to create a map that embeds the observations in a lower dimensional space, while preserving distances or similarities. Different methods vary by their choice of distance or dissimilarity functions, as shown below.

**Diffusion maps** In diffusion maps [5], an embedding in $k$ dimensions is defined via the first $k$ non-trivial right eigenvectors of the t-steps random walk $\boldsymbol{P}^t$ weighted by their eigenvalues. The embedding preserves the ***diffusion distance*** $DM_{\boldsymbol{P}}(x_i, x_j) := \|(\boldsymbol{\delta_i}\boldsymbol{P}^t - \boldsymbol{\delta_j}\boldsymbol{P}^t)(1/\boldsymbol{\pi})\|_2$, where $\boldsymbol{\delta_i}$ is a vector such that $(\boldsymbol{\delta_i})_j = 1$ if $j = i$ and $0$ otherwise, and $\boldsymbol{\pi}$ is the stationary distribution of $\boldsymbol{P}$. Intuitively, $DM_{\boldsymbol{P}}(x_i, x_j)$ considers all the $t$-steps paths between $x_i$ and $x_j$. A larger diffusion time can be seen as a low frequency graph filter, i.e. keeping only information from the low frequency transitions such has the stationary distributions. For this reason, using diffusion with $t > 1$ helps denoising the relationship between observations.

**PHATE** This diffusion-based method preserves the ***potential distance*** [22] $PH_{\boldsymbol{P}} := \| -\log \boldsymbol{\delta_i}\boldsymbol{P}^t + \log \boldsymbol{\delta_j}\boldsymbol{P}^t\|_2$, and justifies this approach using the $\log$ transformation to prevent nearest neighbors from dominating the distances. An alternative approach is suggested using a square root transformation. Part of our contributions is to justify the $\log$ transformation from a geometric point of view. The embedding is defined using multidimensional scaling, which we present below.

**SNE, t-SNE, UMAP** Well-known attraction/repulsion methods such as SNE [11], t-SNE [32], and UMAP [19] define an affinity matrix with entries $p_{ij}$ in the ambient space, and another affinity matrix with entries $q_{ij}$ in the embedded space. To define the embedding, a loss between the two affinity matrices is minimized. Specifically, the loss function is $D_{\mathrm{KL}}(p\|q) := \sum_{ij} p_{ij} \log p_{ij}/q_{ij}$ in SNE and t-SNE, whereas UMAP is equivalent to adding $D_{\mathrm{KL}}(1 - p\|1 - q)$ for unnormalized densities [2]. While these methods preserve affinities, they do not preserve any types of distances in the embedding.

# 4   Heat-Geodesic Embedding

In this section, we present our Heat Geodesic Embedding which is summarized in Alg. 1. We start by introducing the heat-geodesic dissimilarity, then present a robust transformation, and a heuristic to choose the optimal diffusion time. Proofs not present in the main text are given in AppendixA.

We consider the discrete case, where we have a set of $n$ points $\{x_i\}_{i=1}^n =: \boldsymbol{X}$ in a high dimensional Euclidean space $x_i \in \mathbb{R}^d$. From this point cloud, we want to define a map $\phi : \mathbb{R}^d \to \mathbb{R}^k$ that embeds the observation in a lower dimensional space. An important property of our embedding is that we preserve manifold geodesic distances in a low dimensional space.

**Heat-geodesic dissimilarity**   Inspired by Varadhan's formula and the Harnack inequalities, we defined a heat-geodesic dissimilarity based on heat diffusion on graphs. From observations (data-points) in $\mathbb{R}^d$, we define an undirected graph $\mathcal{G}$, and compute its heat kernel $\boldsymbol{H}_t = e^{-t\boldsymbol{L}}$, where $\boldsymbol{L}$ is the combinatorial or symmetrically normalized graph Laplacian (the heat kernel is thus symmetric). Following the inequality (2), we can rearange the terms to isolate the geodesic distance, inspired by this observation, we define the following dissimilarity.

**Definition 4.1.** For a diffusion time $t > 0$ and tunable parameter $\sigma > 0$, we define the **heat-geodesic dissimilarity** between $x_i, x_j \in \boldsymbol{X}$ as

$$d_t(x_i, x_j) := [-4t \log(\boldsymbol{H}_t)_{ij} - \sigma 4t \log(\boldsymbol{V}_t)_{ij}]^{1/2}$$

where $\boldsymbol{H}_t$ is the heat kernel on the graph $\mathcal{G}$, and $(\boldsymbol{V}_t)_{ij} := 2[(\boldsymbol{H}_t)_{ii} + (\boldsymbol{H}_t)_{jj}]^{-1}$.

Here the $\log$ is applied elementwise, and the term $-4t \log(\boldsymbol{H}_t)_{ij}$ corresponds to the geodesic approximation when $t \to 0$ as in Varadhan's formula. In practice one uses a fixed diffusion time $t > 0$, so we add a symmetric volume correction term as in the Harnack inequality, ensuring that $d_t(x_i, x_j)$ is symmetric. From Sec. 2, we have $h_t(x, x) \simeq V(x, \sqrt{t})^{-1}$, and we use the diagonal of $\boldsymbol{H}_t$ to approximate the inverse of the volume. With this volume correction term and $\sigma = 1$, the dissimilarity is such that $d_t(x_i, x_i) = 0$ for all $t > 0$. When $\sigma = 0$ or the manifold has uniform volume growth (as in the constant curvature setting) we show that the heat-geodesic dissimilarity is order preserving:

**Proposition 4.2.** *When $\sigma = 0$ or the manifold has uniform volume growth, i.e. $(\boldsymbol{H}_t)_{ii} = (\boldsymbol{H}_t)_{jj}$, and the heat kernel is pointwise monotonically decreasing w.r.t. a norm $|\cdot|$ in ambient space, we have for triples $x_i, x_j, x_k \in \boldsymbol{X}$ that $|x_i - x_j| > |x_i - x_k|$ implies $d_t(x_i, x_j) > d_t(x_i, x_k)$, i.e. the heat-geodesic dissimilarity is order preserving.*

*Proof.* When $\sigma = 0$ or the manifold has uniform volume growth we need only consider the $-4t \log(\boldsymbol{H}_t)_{ij}$ terms. The assumption of pointwise monotonicity of the heat kernel entails that $|x_i - x_j| > |x_i - x_k|$ implies $\boldsymbol{H}_t(x_i, x_j) < \boldsymbol{H}_t(x_i, x_k)$. We are able to conclude that $-4t \log \boldsymbol{H}_t(x_i, x_j) > -4t \log \boldsymbol{H}_t(x_i, x_k)$ and thus $d_t(x_i, x_j) > d_t(x_i, x_k)$.   $\square$

**Denoising distances with triplet computations**   We note that both diffusion maps and PHATE compute a triplet distance between datapoints, i.e., rather than using the direct diffusion probability between datapoints, they use a distance between corresponding rows of a diffusion operator. In particular, diffusion maps uses Euclidean distance, and PHATE uses an M-divergence. Empirically, we notice that this step acts as a denoiser for distances. We formalize this observation in the following proposition. We note $D_T$ the triplet distance. The triplet distance compares the distances relative to other points. Intuitively, this is a denoising step, since the effect of the noise is spread across the entire set of points. For a reference dissimilarity like the heat-geodesic, it is defined as $D_T(x_i, x_j) := \|d_t(x_i, \cdot) - d_t(x_j, \cdot)\|_2$. For linear perturbations of the form $d_t(x_i, x_j) + \epsilon$, where $\epsilon \in \mathbb{R}$, the effect of $\epsilon$ on $D_T(x_i, x_j)$ is less severe than on $d_t(x_i, x_j)$. Our embedding is based on a linear combination between the heat-geodesic dissimilarity and its triplet distance $(1 - \rho)d_t + \rho D_T$, where $\rho \in [0, 1]$.

**Proposition 4.3.** *Denote the perturbed triplet distance by $\widetilde{D_T}(x_i, x_j) = \|\tilde{d}_t(x_i, \cdot) - \tilde{d}_t(x_j, \cdot)\|_2$ where $\tilde{d}_t(x_i, x_j) := d_t(x_i, x_j) + \epsilon$ and $\tilde{d}_t(x_i, x_k) := d_t(x_i, x_k)$ for $k \neq j$. Then the triplet distance $D_T$ is robust to perturbations , i.e., for all $\epsilon > 0$,*

$$\left( \frac{\widetilde{D_T}(x_i, x_j)}{D_T(x_i, x_j)} \right)^2 \leq \left( \frac{d_t(x_i, x_j) + \epsilon}{d_t(x_i, x_j)} \right)^2.$$

**Optimal diffusion time** Varadhan's formula suggests a small value of diffusion time $t$ to approximate geodesic distance on a manifold. However, in the discrete data setting, geodesics are based on graph constructions, which in turn rely on nearest neighbors. Thus, small $t$ can lead to disconnected graphs. Additionally, increasing $t$ can serve as a way of denoising the kernel (which is often computed from noisy data) as it implements a low-pass filter over the eigenvalues, providing the additional advantage of adding noise tolerance. By computing a sequence of heat kernels $(\boldsymbol{H}_t)_t$ and evaluating their entropy $H(\boldsymbol{H}_t) := -\sum_{ij}(\boldsymbol{H}_t)_{ij}\log(\boldsymbol{H}_t)_{ij}$, we select $t$ with the knee-point method [26] on the function $t \mapsto H(\boldsymbol{H}_t)$. We show in Sec. 6.1 that our heuristic for determining the diffusion time automatically leads to better overall results.

**Weighted MDS** The loss in MDS (eq.5) is usually defined with uniform weights. Here, we optionally weight the loss by the heat kernel. In Sec. 5, we will show how this modification relates our method to the embedding defined by SNE[11]. For $x_i, x_j \in \boldsymbol{X}$, we minimize $(\boldsymbol{H}_t)_{ij}(d_t(x_i, x_j) - \|\phi(x_i) - \phi(x_j)\|_2)^2$. This promotes geodesic preservation of local neighbors, since more weights are given to points with higher affinities.

**Heat-geodesic embedding** To define a lower dimensional embedding of a point cloud $\boldsymbol{X}$, we construct a matrix from the heat-geodesic dissimilarity, and then use MDS to create the embedding. Our embedding defines a map $\phi$ that minimizes $\left(d_t(x_i, x_j) - \|\phi(x_i) - \phi(x_j)\|_2\right)^2$, for all $x_i, x_j \in \boldsymbol{X}$. Hence, it preserves the heat-geodesic dissimilarity as the loss decreases to zero. In Alg. 1, we present the main steps of our algorithm using the heat-geodesic dissimilarity. A detailed version is presented in Appendix A.

---

**Algorithm 1** Heat Geodesic Embedding

---

1: **Input:** $N \times d$ dataset matrix $\boldsymbol{X}$, denoising parameter $\rho \in [0, 1]$, Harnack regularization $\sigma > 0$, output dimension $k$.
2: **Returns:** $N \times k$ embedding matrix $\boldsymbol{E}$.
3: $\boldsymbol{H}_t \leftarrow p_K(\boldsymbol{L}, t)$           ▷ *Heat approximation*
4: $t \leftarrow \text{Kneedle}\{H(\boldsymbol{H}_t)\}_t$        ▷ *Knee detection e.g. [26]*
5: $\boldsymbol{D} \leftarrow [-4t\log(\boldsymbol{H}_t)_{ij} - \sigma 4t\log(\boldsymbol{V}_t)_{ij}]^{1/2}$    ▷ $\log$ *is applied elementwise*
6: $\boldsymbol{D} \leftarrow (1 - \rho)\boldsymbol{D} + \rho D_{\mathrm{T}}$        ▷ *Triplet interpolation step*
7: Return $\boldsymbol{E} \leftarrow \text{MetricMDS}(\boldsymbol{D}, \|\cdot\|_2, k)$

---

## 5 Relation to other manifold learning methods

In this section, we elucidate theoretical connections between the Heat Geodesic Embedding and other manifold learning methods. We relate embeddings via the eigenvalues of $\boldsymbol{H}_t$ or $\boldsymbol{P}^t$ with Laplacian eigenmaps and diffusion maps. We then present the relation between our methods and PHATE and SNE. We provide further analysis in the Appendix A. In particular, we introduce a new definition of kernel preserving embeddings; either via kernel-based distances (diffusion maps, PHATE) or via similarities (e.g. t-SNE, UMAP).

**Diffusion maps with the heat kernel** Diffusion maps [5] define an embedding with the first $k$ eigenvectors $(\phi_i)_i$ of $\boldsymbol{P}$, while Laplacian eigenmaps [1] uses the eigenvectors $(\psi_i)_i$ of $\boldsymbol{L}$. In the following, we recall the links between the two methods, and show that a rescaled Laplacian eigenmaps preserves the diffusion distance with the heat kernel $\boldsymbol{H}_t$.

**Lemma 5.1.** *Rescaling the Laplacian eigenmaps embedding with $x_i \mapsto (e^{-2t\lambda_1}\psi_{1,i}, \ldots, e^{-2t\lambda_k}\psi_{k,i})$ preserves the diffusion distance $DM_{\boldsymbol{H}_t}$.*

**Relation to PHATE** The potential distance in PHATE (Sec. 3) is defined by comparing the transition probabilities of two $t$-steps random walks initialized from different vertices. The transition matrix $\boldsymbol{P}^t$ mimics the heat propagation on a graph. The heat-geodesic dissimilarity provides a new interpretation of PHATE. In the following proposition, we show how the heat-geodesic relates to the PHATE potential distance with a linear combination of $t$-steps random walks.

**Proposition 5.2.** *The PHATE potential distance with the heat kernel $PH_{\boldsymbol{H}_t}$ can be expressed in terms of the heat-geodesic dissimilarity with $\sigma = 0$*

$$PH_{\boldsymbol{H}_t} = (1/4t)^2 \|d_t(x_i, \cdot) - d_t(x_j, \cdot)\|_2^2,$$

*and it is equivalent to a multiscale random walk distance with kernel $\sum_{k>0} m_t(k)\boldsymbol{P}^k$, where $m_t(k) := t^k e^{-t}/k!$.*

*Proof.* We present a simplified version of the proof, more details are available in Appendix A. For $\sigma = 0$, we have $d_t(x_i, x_j) = -4t \log(\boldsymbol{H}_t)_{ij}$, the relation between the PHATE potential and the heat-geodesic follows from the definition

$$PH_{\boldsymbol{H}_t}(x_i, x_j) = \sum_k \big( -\log \boldsymbol{H}_t(x_i, x_k) + \log \boldsymbol{H}_t(x_j, x_k) \big)^2 = (1/4t)^2 \|d_t(x_i, \cdot) - d_t(x_j, \cdot)\|_2^2.$$

Using the heat kernel $\boldsymbol{H}_t$ with the random walk Laplacian $\boldsymbol{L}_{rw} = \boldsymbol{Q}^{-1}\boldsymbol{L} = \boldsymbol{I}_n - \boldsymbol{Q}^{-1}\boldsymbol{W}$ corresponds to a multiscale random walk kernel. We can write $\boldsymbol{L}_{rw} = \boldsymbol{S}\Lambda\boldsymbol{S}^{-1}$, where $\boldsymbol{S} := \boldsymbol{Q}^{-1/2}\Psi$. Since $\boldsymbol{P} = \boldsymbol{I}_n - \boldsymbol{L}_{rw}$, we have $\boldsymbol{P}^t = \boldsymbol{S}(\boldsymbol{I}_n - \Lambda)^t\boldsymbol{S}^{-1}$. Interestingly, we can relate the eigenvalues of $\boldsymbol{H}_t$ and $\boldsymbol{P}$ with the Poisson distribution. The probability mass function of a Poisson distribution with mean $t$ is given by $m_t(k) := t^k e^{-t}/k!$. For $t \geq 0$, we have $e^{-t(1-\mu)} = \sum_{k\geq 0} m_t(k)\mu^k$. With this relationship, we can express $\boldsymbol{H}_t$ as a linear combination of $\boldsymbol{P}^t$ weighted by the Poisson distribution. Indeed, substituting $\lambda = 1 - \mu$ in yields

$$\boldsymbol{H}_t = \boldsymbol{S}e^{-t\Lambda}\boldsymbol{S}^{-1} = \boldsymbol{S}\sum_{k=0}^{\infty} m_t(k)(\boldsymbol{I}_n - \Lambda)^k\boldsymbol{S}^{-1} = \sum_{k=0}^{\infty} m_t(k)\boldsymbol{P}^k.$$

$\square$

*Remark* 5.3. In the previous proposition, the same argument holds for the symmetric Laplacian and the affinity matrix $\boldsymbol{A} := \boldsymbol{Q}^{-1/2}\boldsymbol{W}\boldsymbol{Q}^{-1/2}$ used in other methods such as diffusion maps [5]. This is valid since we can write $\boldsymbol{L}_{sym} = \boldsymbol{Q}^{-1/2}\Psi\Lambda\Psi^T\boldsymbol{Q}^{-1/2}$, and $\boldsymbol{A} = \boldsymbol{I}_n - \boldsymbol{L}_{sym}$.

*Remark* 5.4. This proposition shows that, as the denoising parameter $\rho \to 1$, Heat Geodesic Embedding interpolates to the PHATE embeddings with a weighted kernel $\sum_{k=0}^{\infty} m_t(k)\boldsymbol{P}^k$.

**Relation to SNE** The heat-geodesic method also relates to SNE [11], and its variation using the Student distribution t-SNE [18]. In SNE, the similarity between points is encoded via transition probabilities $p_{ij}$. The objective is to learn an affinity measure $q$, that depends on the embedding distances $\|y_i - y_j\|_2$, such that it minimizes $D_{\mathrm{KL}}(p\|q)$. Intuitively, points that have a strong affinity in the ambient space, should also have a strong affinity in the embedded space. Even though the heat-geodesic minimization is directly on the embedding distances, we can show an equivalent with SNE. In Appendix A, we provide additional comparisons between SNE and our method.

**Proposition 5.5.** *The Heat-Geodesic embedding with $\sigma = 0$ and squared distances minimization weighted by the heat kernel is equivalent to SNE with the heat kernel affinity in the ambient space, and a Gaussian kernel in the embedded space $q_{ij} = \exp(-\|y_i - y_j\|^2/4t)$.*

# 6   Results

In this section, we show the versatility of our method, showcasing its performance in terms of clustering and preserving the structure of continuous manifolds. We compare the performance of Heat Geodesic Embedding with multiple state-of-the-art baselines on synthetic datasets and real-world datasets. For all models, we perform sample splitting with a 50/50 validation-test split. The validation and test sets each consists of 5 repetitions with different random initializations. The hyper-parameters are selected according to the performance on the validation set. We always report the results on the test set, along with the standard deviations computed over the five repetitions. We use the following methods in our experiments: our *Heat Geodesic Embedding*, *diffusion maps* [5], *PHATE* [22], *shortest-path* (used in Isomap [31]) which estimates the geodesic distance by computing the shortest path between two nodes in a graph built on the point clouds, *t-SNE* [32], *UMAP* [19], and metric MDS with Euclidean distance. Details about each of these methods, and results for different parameters (graph type, heat approximation, etc.) are given in Appendix C.

Table 1: Pearson and Spearman correlation between the inferred and ground truth distance matrices on the Swiss roll and Tree datasets (higher is better). Best models on average are bolded.

|  | Swiss roll | | Tree | |
| --- | --- | --- | --- | --- |
| Distance | Pearson | Spearman | Pearson | Spearman |
| Diffusion distance | $0.476 \pm 0.226$ | $0.478 \pm 0.138$ | $0.656 \pm 0.054$ | $0.653 \pm 0.057$ |
| PHATE potential | $0.457 \pm 0.01$ | $0.404 \pm 0.024$ | $0.766 \pm 0.023$ | $0.743 \pm 0.028$ |
| Shortest path | $0.497 \pm 0.144$ | $0.558 \pm 0.134$ | $0.780 \pm 0.009$ | $0.757 \pm 0.019$ |
| Euclidean | $0.365 \pm 0.006$ | $0.413 \pm 0.005$ | $0.735 \pm 0.014$ | $0.704 \pm 0.033$ |
| Heat-geodesic (ours) | $\mathbf{0.702 \pm 0.086}$ | $\mathbf{0.700 \pm 0.073}$ | $\mathbf{0.822 \pm 0.008}$ | $\mathbf{0.807 \pm 0.016}$ |

## 6.1 Distance matrix comparison

We start by evaluating the ability of the different distances or dissimilarities to recover the ground truth distance matrix of a point cloud. For this task, we use the Swiss roll and Tree datasets, for which the ground truth geodesic distance is known. The Swiss roll dataset consists of data points sampled on a smooth manifold (see Fig. 1). The Tree dataset is created by connecting multiple high-dimensional Brownian motions in a tree-shape structure. In Fig. 1, we present embeddings of both datasets. Our method recovers the underlying geometry, while other methods create artificial clusters or have too much denoising. Because we aim at a faithful relative distance between data points, we compare the methods according to the Pearson and Spearman correlations of the estimated distance matrices with respect to ground truth. Results are displayed in Tab. 1. We observe that Heat Geodesic Embedding typically outperforms previous methods in terms of the correlation with the ground truth distance matrix, confirming the theoretical guarantees provided in Sec. 4 & 2. Additional results such as computation time and correlation for different noise levels are available in Appendix C.

**Optimal diffusion time** In Section 4, we described a heuristic to automatically choose the diffusion time based on the entropy of $\boldsymbol{H}_t$. In Fig. 2, we show that the knee-point of $t \mapsto H(\boldsymbol{H}_t)$, corresponds to a high correlation with the ground distance, while yielding a low approximation error of the distance matrix (measured by the Frobenius norm of the difference between $\boldsymbol{D}$ and the ground truth).

## 6.2 Preservation of the inherent data structure

A crucial evaluation criteria of manifold learning methods is the ability to capture the inherent structure of the data. For instance, clusters in the data should be visible in the resulting low dimensional representation. Similarly, when the dataset consists of samples taken at different time points, one expects to be able to characterize this temporal evolution in the low dimensional embedding [22]. We thus compare the different embedding methods according to their ability to retain clusters and temporal evolution of the data.

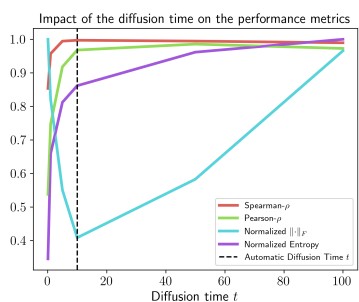

Figure 2: Evolution of the correlation between estimated and ground truth distance matrices in function of the diffusion time $t$.

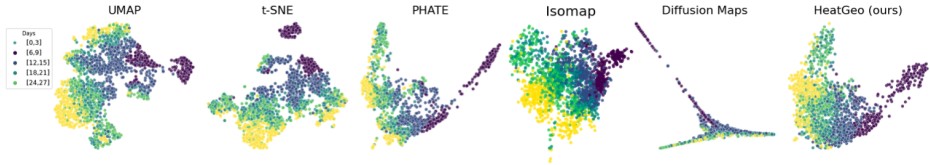

Figure 3: Embeddings of 2000 differentiating cells from embryoid body [22] over 28 days. UMAP and t-SNE do not capture the continuous manifold representing the cells' evolution.

**Identifying clusters.** We use the PBMC dataset, the Swiss roll, the Tree dataset, MNIST [8], and COIL-20 [23] dataset. The PBMC dataset consists of single-cell gene expressions from 3000 individual peripheral blood mononuclear cells. Cells are naturally clustered by their cell type. For

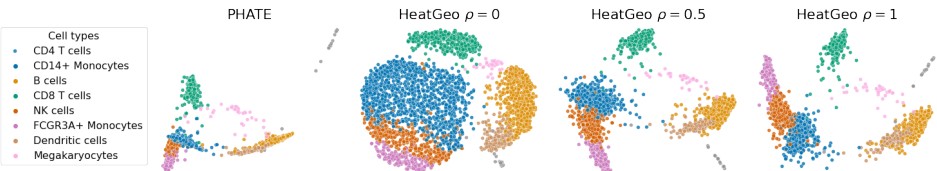

Figure 4: Embeddings on PBMC using the triplet distance with the heat-geodesic for different regularization parameter $\rho$.

the Tree dataset, we use the branches as clusters. For the Swiss roll dataset, we sample data points on the manifold according to a mixture of Gaussians and use the mixture component as the ground truth cluster labels. The MNIST and COIL-20 datasets are clustered by digits or objects but may not respect the manifold hypothesis. For each method, we run k-means on the two-dimensional embedding and compare the resulting cluster assignments with ground truth. Tab. 10 reports the results in terms of homogeneity and adjusted mutual information (aMI). Heat Geodesic Embedding is competitive with PHATE and outperforms t-SNE and UMAP on all metrics except on the MNIST and COIL-20 datasets. Yet, we show in Appendix C that all methods tend to perform equally well when the noise level increases. In Fig. 4, we present the PBMC embeddings of PHATE and HeatGeo, showing that HeatGeo interpolates to PHATE for $\rho \to 1$.

Table 2: Clustering quality metrics for different methods. We report the homogeneity and the adjusted mutual information (aMI). Best models on average are bolded (higher is better).

| Method | Swiss roll Homogeneity | Swiss roll aMI | Tree Homogeneity | Tree aMI | PBMC Homogeneity | PBMC aMI | MNIST (Non Manifold) Homogeneity | MNIST (Non Manifold) aMI | COIL (Non Manifold) Homogeneity | COIL (Non Manifold) aMI |
|---|---|---|---|---|---|---|---|---|---|---|
| UMAP | $0.810 \pm 0.036$ | $0.726 \pm 0.045$ | $0.678 \pm 0.086$ | $0.681 \pm 0.086$ | $0.177 \pm 0.037$ | $0.148 \pm 0.035$ | $0.851 \pm 0.016$ | $0.86 \pm 0.015$ | $0.871 \pm 0.009$ | $0.826 \pm 0.012$ |
| t-SNE | $0.748 \pm 0.067$ | $0.668 \pm 0.068$ | $0.706 \pm 0.054$ | $0.712 \pm 0.055$ | $0.605 \pm 0.019$ | $0.544 \pm 0.022$ | $\mathbf{0.903 \pm 0.003}$ | $\mathbf{0.902 \pm 0.003}$ | $\mathbf{0.907 \pm 0.014}$ | $\mathbf{0.88 \pm 0.02}$ |
| Isomap | $0.806 \pm 0.089$ | $0.743 \pm 0.065$ | $0.673 \pm 0.045$ | $0.691 \pm 0.042$ | $0.242 \pm 0.007$ | $0.21 \pm 0.007$ | $0.742 \pm 0.001$ | $0.74 \pm 0.001$ | / | / |
| Non-Metric MDS | $0.003 \pm 0.001$ | $0.0 \pm 0.001$ | $0.003 \pm 0.001$ | $0.001 \pm 0.001$ | $0.011 \pm 0.003$ | $0.001 \pm 0.003$ | $0.019 \pm 0.003$ | $0.001 \pm 0.004$ | $0.296 \pm 0.02$ | $0.005 \pm 0.027$ |
| PHATE | $0.731 \pm 0.035$ | $0.652 \pm 0.046$ | $0.550 \pm 0.042$ | $0.555 \pm 0.042$ | $\mathbf{0.798 \pm 0.012}$ | $\mathbf{0.785 \pm 0.01}$ | $0.822 \pm 0.01$ | $0.835 \pm 0.011$ | $0.804 \pm 0.017$ | $0.735 \pm 0.021$ |
| Diffusion Maps | $0.643 \pm 0.053$ | $0.585 \pm 0.051$ | $0.341 \pm 0.103$ | $0.358 \pm 0.093$ | $0.026 \pm 0.001$ | $0.038 \pm 0.001$ | $0.556 \pm 0.002$ | $0.622 \pm 0.002$ | $0.21 \pm 0.036$ | $0.142 \pm 0.024$ |
| HeatGeo (ours) | $\mathbf{0.820 \pm 0.008}$ | $\mathbf{0.740 \pm 0.018}$ | $\mathbf{0.784 \pm 0.051}$ | $\mathbf{0.786 \pm 0.051}$ | $0.734 \pm 0.009$ | $0.768 \pm 0.017$ | $0.785 \pm 0.0$ | $0.829 \pm 0.001$ | $0.849 \pm 0.016$ | $0.806 \pm 0.022$ |

**Temporal data representation.** For this task, we aim at representing data points from population observed at consecutive points in time. We use single cell gene expression datasets collected across different time points, including the Embryoid Body (EB), IPSC [22], and two from the 2022 NeurIPS multimodal single-cell integration challenge (Cite & Multi). To quantitatively evaluate the quality of the continuous embeddings, we first embed the entire dataset and obfuscate all samples from a particular time point (*e.g.,* $t = 2$). We then estimate the distribution of the missing time point by using displacement interpolation [35] between the adjacent time points (*e.g.,* $t = 1$ and $t = 3$). We report the Earth Mover Distance (EMD) between the predicted distribution and true distribution. A low EMD suggests that the obfuscated embeddings are naturally located between the previous and later time points, and that the generated embedding captures the temporal evolution of the data adequately. Results are presented in Tab. 3. Heat Geodesic Embedding outperforms other methods on the EB, Multi, and IPSC datasets and is competitive with other approaches on Cite. We show a graphical depiction of the different embeddings for the embryoid (EB) dataset in Fig. 3.

Table 3: EMD between a linear interpolation of two consecutive time points $t - 1$, $t + 1$, and the time points $t$. Best models on average are bolded (lower is better).

| Method | Cite | EB | Multi | IPSC |
|---|---|---|---|---|
| Euclidean | $0.978 \pm 0.069$ | $1.012 \pm 0.039$ | $1.212 \pm 0.199$ | $1.085 \pm 0.234$ |
| Isomap | $0.978 \pm 0.105$ | $0.993 \pm 0.062$ | $1.299 \pm 0.307$ | $1.026 \pm 0.253$ |
| Non-metric MDS | $0.81 \pm 0.012$ | $0.85 \pm 0.014$ | $0.806 \pm 0.015$ | $1.013 \pm 0.067$ |
| UMAP | $\mathbf{0.791 \pm 0.045}$ | $0.942 \pm 0.053$ | $1.418 \pm 0.042$ | $0.866 \pm 0.058$ |
| t-SNE | $0.905 \pm 0.034$ | $0.964 \pm 0.032$ | $1.208 \pm 0.087$ | $1.006 \pm 0.026$ |
| PHATE | $1.032 \pm 0.037$ | $1.088 \pm 0.012$ | $1.254 \pm 0.042$ | $0.955 \pm 0.033$ |
| Diffusion Maps | $0.989 \pm 0.080$ | $0.965 \pm 0.077$ | $1.227 \pm 0.086$ | $0.821 \pm 0.039$ |
| HeatGeo (ours) | $0.890 \pm 0.046$ | $\mathbf{0.733 \pm 0.036}$ | $\mathbf{0.958 \pm 0.044}$ | $\mathbf{0.365 \pm 0.056}$ |

# 7 Conclusion and Limitations

The ability to visualize complex high-dimensional data in an interpretable and rigorous way is a crucial tool of scientific discovery. In this work, we took a step in that direction by proposing a general framework for understanding diffusion-based dimensionality reduction methods through the lens of Riemannian geometry. This allowed us to define a novel embedding based on the heat geodesic dissimilarity—a more direct measure of manifold distance. Theoretically, we showed that our methods brings greater versatility than previous approaches and can help gaining insight into popular manifold learning methods such as diffusion maps, PHATE, and SNE. Experimentally, we demonstrated that it also results in better geodesic distance preservation and excels both at clustering and preserving the structure of a continuous manifold. This contrasts with previous methods that are typically only effective at a single of these tasks.

Despite the strong theoretical and empirical properties, our work presents some limitations. For instance, our method is based on a similarity measure, which is a relaxation of a distance metric. Additionally, the Harnack equation suggests that our parameters for the volume correction could be tuned depending on the underlying manifold. We envision that further analysis of this regularization is a fruitful direction for future work.

## Acknowledgments and Disclosure of Funding

This research was enabled in part by compute resources provided by Mila (mila.quebec). It was partially funded and supported by ESP *Mérite* [G.H.], CIFAR AI Chair [G.W.], NSERC Discovery grant 03267 [G.W.], NIH grants (1F30AI157270-01, R01HD100035, R01GM130847, R01GM135929) [G.W.,S.K.], NSF Career grant 2047856 [S.K.], the Chan-Zuckerberg Initiative grants CZF2019-182702 and CZF2019-002440 [S.K.], the Sloan Fellowship FG-2021-15883 [S.K.], and the Novo Nordisk grant GR112933 [S.K.]. The content provided here is solely the responsibility of the authors and does not necessarily represent the official views of the funding agencies. The funders had no role in study design, data collection & analysis, decision to publish, or preparation of the manuscript.

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

# Appendix

## A  Theory and algorithm details

### A.1  Kernel preserving embeddings

In this section, we attempt to create a generalized framework for dimensionality reduction methods. These methods often have been viewed as disparate or competing but here we show that many of them are related to one another given the right template for methodology comparison. In order to do this, we introduce a general definition suited for distance-preserving dimensionality reduction methods. With this definition, we can cast many dimensionality reduction methods within the same framework, and easily compare them. We recall that the observations in the ambient space are denoted $x$, and those in the embedded space are denoted $y$. The definition relies on kernel functions $H_t^x$, $H_t^y$ defined respectively on the ambient and embedded spaces and on transformations $T^x$, $T^y$ applied to the kernels. We recall that a divergence $f : \mathbb{R} \times \mathbb{R} \to \mathbb{R}^+$ is such that $f(a, b) = 0$ if and only if $a = b$ and $f(a, a + \delta)$ is a positive semi-definite quadratic form for infinitesimal $\delta$.

**Definition A.1.** We define a **kernel features preserving embedding** as an embedding which minimizes a loss $L$ between a transformation $T^x$ of the ambient space kernel $H_t^x$ and its embedded space counterpart

$$L := f(T^x(H_t^x), T^y(H_{t'}^y)), \tag{6}$$

where $f$ is any $C^2$ divergence on $\mathbb{R}_{\geq 0}$.

*Example* 1. We formulate MDS as a kernel feature-preserving embedding. Suppose we want to preserve the Euclidean distance, we have $H_t^x(x_i, x_j) = \|x_i - x_j\|_2$, $H_t^y(y_i, y_j) = \|y_i - y_j\|_2$, $f(a, b) = \|a - b\|_2$, and $T^x = T^y = I$.

In the following, we present popular dimensionality reduction methods that are kernel features preserving embeddings. With this definition, we can distinguish between methods that a preserve a kernel via affinities or distances. For the methods considered in this work, $H_t^x$ is an affinity kernel, but its construction varies from one method to another. In PHATE and Diffusion maps, $H_t^x$ is a random walk $\boldsymbol{P}$, while in Heat Geodesic Embedding we use the heat kernel $\boldsymbol{H}_t$. t-SNE defines $H_t^x$ as a symmetrized random walk matrix from a Gaussian kernel, while UMAP uses an unnormalized version. Methods such as PHATE and diffusion maps define a new distance matrix from a kernel in the ambient space and preserve these distances in the embedded space. Other methods like t-SNE and UMAP define similarities from a kernel and aim to preserve these similarities in the ambient space and embedded space via an entropy-based loss. We note the Kullback–Leibler divergence $D_{\mathrm{KL}}(a, b) = \sum_{ij} a_{ij} \log[a_{ij}/b_{ij}]$.

**Proposition A.2.** *The embeddings methods HeatGeo, PHATE, Diffusion Maps, SNE, t-SNE, and UMAP are kernel feature-preserving embeddings.*

*Proof.* We assume that the affinity kernel in the ambient space $H_t^x$, is given, to complete the proof we need to define $f, H_t^y, T^x, T^y$ for all methods.

We start with the distance preserving embeddings; HeatGeo, PHATE, and Diffusion Maps. For these methods, the kernel in the embed space is simply $H_t^y(y_i, y_j) = \|y_i - y_j\|_2$, without transformation, i.e. $T^y = I$. Since they preserve a distance, the loss is $f(T^x(H_t^x), T^y(H_{t'}^y)) = \|H_t^x - H_{t'}^y\|_2$.

In the Heat Geodesic Embedding we apply a transformation on $H_t^x = \boldsymbol{H}_t$ to define a dissimilarity, hence $T^x(H_t^x) = -t \log H_t^x$ (for $\sigma = 0$), where $\log$ is applied elementwise.

In PHATE, the potential distance is equivalent to $(T^x(H_t^x))_{ij} = \| - \log(H_t^x)_i + \log(H_t^x)_j\|_2$. In Diffusion Maps, the diffusion distance is $(T^x(H_t^x))_{ij} = \|(H_t^x)_i - (H_t^x)_j\|_2$.

SNE, t-SNE, and UMAP preserve affinities from a kernel. For these three methods, the loss is a divergence between distributions, namely $f = D_{\mathrm{KL}}$. They vary by defining different affinity kernel and transformation in the embedded space. SNE uses the unnormalized kernel $H_t^y(y_i, y_j) = \exp(-(1/t)\|y_i - y_j\|_2^2)$, with $T^x = T^y = I$. Whereas, t-SNE uses $(H_1^y)_{ij} = (1 + \|y_i - y_j\|^2)^{-1}$,

Table 4: Overview of kernel preserving methods.

| Method | $H_t^y(y_i, y_j)$ | $T^x(H_t^x)$ | $T^y(H_t^y)$ | $f$ |
|---|---|---|---|---|
| PHATE | $\|y_i - y_j\|_2$ | $\| -\log(H_t^x)_i + \log(H_t^x)_j \|_2$ | $H_t^y$ | $\|\cdot\|_2$ |
| Heat Geodesic | $\|y_i - y_j\|_2$ | $-t\log(H_t^x)_{ij}$ | $H_t^y$ | $\|\cdot\|_2$ |
| Diffusion Maps | $\|y_i - y_j\|_2$ | $\|(H_t^x)_i - (H_t^x)_j\|_2$ | $H_t^y$ | $\|\cdot\|_2$ |
| SNE | $\exp(-(\frac{1}{t})\|y_i - y_j\|_2^2)$ | $H_t^x$ | $H_t^y$ | $D_{\mathrm{KL}}$ |
| t-SNE | $(1 + \|y_i - y_j\|^2)^{-1}$ | $H_t^x$ | $H_t^y$ | $D_{\mathrm{KL}}$ |
| UMAP | $(1 + \|y_i - y_j\|^2)^{-1}$ | $H_t^x$ | $\frac{(H_1^y)_{ij}}{(1-(H_1^y)_{ij})}$ | $D_{\mathrm{KL}}$ |

and $T^x = T^y = I$. UMAP define a pointwise transformation in the embedded space with $(H_1^y)_{ij} = (1 + \|y_i - y_j\|^2)^{-1}$, $(T^y(H_t^y))_{ij} = (H_1^y)_{ij}/(1 - (H_1^y)_{ij})$, and $T^x = I$.

We summarize the choice of kernels and functions in Tab. 4 $\qquad\square$

### A.2 Proofs

**Proposition 4.3.** *Denote the perturbed triplet distance by* $\widetilde{D_{\mathrm{T}}}(x_i, x_j) = \|\tilde{d}_t(x_i, \cdot) - \tilde{d}_t(x_j, \cdot)\|_2$ *where* $\tilde{d}_t(x_i, x_j) := d_t(x_i, x_j) + \epsilon$ *and* $\tilde{d}_t(x_i, x_k) := d_t(x_i, x_k)$ *for* $k \neq j$. *Then the triplet distance* $D_{\mathrm{T}}$ *is robust to perturbations , i.e., for all* $\epsilon > 0$,

$$\left( \frac{\widetilde{D_{\mathrm{T}}}(x_i, x_j)}{D_{\mathrm{T}}(x_i, x_j)} \right)^2 \leq \left( \frac{d_t(x_i, x_j) + \epsilon}{d_t(x_i, x_j)} \right)^2 .$$

*Proof of Proposition 4.3.* The effect of the noise on the square distance is $(d_t(x_i, x_j) + \epsilon)^2/d(x_i, x_j)^2 = 1 + (2\epsilon d_t(x_i, x_j) + \epsilon^2)/d(x_i, x_j)^2$. Denoting the perturbed triplet distance by $\widetilde{D_{\mathrm{T}}}$, we have

$$\frac{\widetilde{D_{\mathrm{T}}}(x_i, x_j)^2}{D_{\mathrm{T}}(x_i, x_j)^2} = \frac{\sum_{k \neq i,j} \left( d_t(x_i, x_k) - d_t(x_j, x_k) \right)^2 + 2(d_t(x_i, x_j) + \epsilon)^2}{D_{\mathrm{T}}(x_i, x_j)^2} = 1 + \frac{4\epsilon d(x_i, x_j) + 2\epsilon^2}{D_{\mathrm{T}}(x_i, x_j)^2},$$

and we have

$$\frac{4\epsilon d(x_i, x_j) + 2\epsilon^2}{D_T(x_i, x_j)^2} \leq \frac{2\epsilon d_t(x_i, x_j) + \epsilon^2}{d_t(x_i, x_j)^2}$$

For $\epsilon > 0$, this gives

$$\epsilon \geq \frac{4d_t(x_i, x_j)^3 - 2d_t(x_i, x_j)D_T(x_i, x_j)^2}{D_t(x_i, x_j)^2 - 2d_t(x_i, x_j)^2} = -2d_t(x_i, x_j).$$

For $\epsilon < 0$, we have

$$\epsilon \leq \frac{4d_t(x_i, x_j)^3 - 2d_t(x_i, x_j)D_T(x_i, x_j)^2}{D_t(x_i, x_j)^2 - 2d_t(x_i, x_j)^2} = -2d_t(x_i, x_j).$$

Thus $\epsilon \in (-\infty, -2d_t(x_i, x_j)) \cup (0, \infty)$. As we require the perturbation factor $\epsilon \ll d_t(x_i, x_j)$, hence we choose $\epsilon \in (0, \infty)$.

$\qquad\square$

**Lemma 5.1.** *Rescaling the Laplacian eigenmaps embedding with* $x_i \mapsto (e^{-2t\lambda_1}\psi_{1,i}, \ldots, e^{-2t\lambda_k}\psi_{k,i})$ *preserves the diffusion distance* $DM_{\boldsymbol{H}_t}$.

*Proof of Lemma 5.1.* Since the eigendecompotision of $\boldsymbol{H}_t$ form an orthonormal basis of $\mathbb{R}^n$, and since its first eigenvector is constant, we can write the diffusion distance $\|\boldsymbol{\delta_i H}_t - \boldsymbol{\delta_i H}_t\|_2^2 = \sum_{k \geq 0} e^{-2t\lambda_k}(\psi_{ki} - \psi_{kj})^2 = \sum_{k \geq 1} e^{-2t\lambda_k}(\psi_{ki} - \psi_{kj})^2$. In particular, this defines the $k$ dimensional embedding $x \mapsto (e^{-t\lambda_1}\psi_1(x), \ldots, e^{-t\lambda_k}\psi_k(x))$. $\qquad\square$

**Proposition 5.2.** *The PHATE potential distance with the heat kernel $PH_{\boldsymbol{H}_t}$ can be expressed in terms of the heat-geodesic dissimilarity with $\sigma = 0$*

$$PH_{\boldsymbol{H}_t} = (1/4t)^2 \|d_t(x_i, \cdot) - d_t(x_j, \cdot)\|_2^2,$$

*and it is equivalent to a multiscale random walk distance with kernel $\sum_{k>0} m_t(k)\boldsymbol{P}^k$, where $m_t(k) := t^k e^{-t}/k!$.*

*Proof of Proposition 5.2.* For $\sigma = 0$, we have $d_t(x_i, x_j) = -4t \log(\boldsymbol{H}_t)_{ij}$, the relation between the PHATE potential and the heat-geodesic follows from the definition

$$PH_{\boldsymbol{H}_t} = \sum_k \big( -\log \boldsymbol{H}_t(x_i, x_k) + \log \boldsymbol{H}_t(x_j, x_k) \big)^2$$

$$= (1/4t)^2 \|d_t(x_i, \cdot) - d_t(x_j, \cdot)\|_2^2.$$

Using the heat kernel $\boldsymbol{H}_t$ with the random walk Laplacian $\boldsymbol{L}_{rw} = \boldsymbol{Q}^{-1}\boldsymbol{L} = \boldsymbol{I}_n - \boldsymbol{Q}^{-1}\boldsymbol{W}$ corresponds to a multiscale random walk kernel. Recall that we can write $\boldsymbol{L}_{rw}$ in terms of the symmetric Laplacian $\boldsymbol{L}_{rw} = \boldsymbol{Q}^{-1/2}\boldsymbol{L}_s\boldsymbol{Q}^{1/2}$, meaning that the two matrices are similar, hence admit the same eigenvalues $\Lambda$. We also know that $\boldsymbol{L}_s$ is diagonalizable, since we can write $\boldsymbol{L}_s = \boldsymbol{Q}^{-1/2}\boldsymbol{L}\boldsymbol{Q}^{-1/2} = \boldsymbol{Q}^{-1/2}\Psi\Lambda\Psi^T\boldsymbol{Q}^{-1/2}$. In particular, we have $\boldsymbol{L}_{rw} = \boldsymbol{S}\Lambda\boldsymbol{S}^{-1}$, where $\boldsymbol{S} := \boldsymbol{Q}^{-1/2}\Psi$. The random walk matrix can be written as $\boldsymbol{P} = \boldsymbol{I}_n - \boldsymbol{R}_{rw}$, hence its eigenvalues are $(\boldsymbol{I}_n - \Lambda)$, and we can write $\boldsymbol{P}^t = \boldsymbol{S}(\boldsymbol{I}_n - \Lambda)^t\boldsymbol{S}^{-1}$. Similarly, the heat kernel with the random walk Laplacian can be written as $\boldsymbol{H}_t = \boldsymbol{S}e^{-t\Lambda}\boldsymbol{S}^{-1}$. Interestingly, we can relate the eigenvalues of $\boldsymbol{H}_t$ and $\boldsymbol{P}$ with the Poisson distribution. Note the probability mass function of a Poisson as $m_t(k) := t^k e^{-t}/k!$, for $t \geq 0$, we have

$$e^{-t(1-\mu)} = e^{-t} \sum_{k\geq 0} \frac{(t\mu)^k}{k!} = \sum_{k\geq 0} m_t(k)\mu^k. \tag{7}$$

We note that (7) is the probability generating function of a Poisson distribution with parameter $t$, i.e. $\mathbb{E}[\mu^X]$, where $X \sim \text{Poisson}(t)$. With this relationship, we can express $\boldsymbol{H}_t$ as a linear combination of $\boldsymbol{P}^t$ weighted by the Poisson distribution. Indeed, substituting $\lambda = 1 - \mu$ in (7) links the eigenvalues of $\boldsymbol{H}_t$ and $\boldsymbol{P}$. We write the heat kernel as a linear combination of random walks weighted by the Poisson distribution, we have

$$\boldsymbol{H}_t = \boldsymbol{S}e^{-t\Lambda}\boldsymbol{S}^{-1} = \boldsymbol{S}\sum_{k=0}^{\infty} m_t(k)(\boldsymbol{I}_n - \Lambda)^k\boldsymbol{S}^{-1} = \sum_{k=0}^{\infty} m_t(k)\boldsymbol{P}^k.$$

$\square$

**Proposition 5.5.** *The Heat-Geodesic embedding with $\sigma = 0$ and squared distances minimization weighted by the heat kernel is equivalent to SNE with the heat kernel affinity in the ambient space, and a Gaussian kernel in the embedded space $q_{ij} = \exp(-\|y_i - y_j\|^2/4t)$.*

*Proof of Proposition 5.5.* The squared MDS weighted by the heat kernel corresponds to

$$\sum_{ij} h_t(x_i, x_j)(d_{ij}^2 - \|y_i - y_j\|^2)^2 = \sum_{ij} h_t(x_i, x_j)(-t \log h_t(x_i, x_j) - \|y_i - y_j\|^2)^2$$

$$= \sum_{ij} h_t(x_i, x_j)t^2(\log h_t(x_i, x_j) - \log \exp(-\|y_i - y_j\|^2/t)^2.$$

If there exists an embedding that attain a zero loss, then it is the same as $\sum_{ij} h_t(x_i, x_j)(\log h_t(x_i, x_j) - \log \exp(-\|y_i - y_j\|^2/t) = D_{\text{KL}}(h_t\|q)$. $\square$

### A.3 Algorithm details

We present a detailed version of the Heat Geodesic Embedding algorithm in Alg.2.

For the knee-point detection we use the Kneedle algorithm [26]. It identifies a knee-point as a point where the curvature decreases maximally between points (using finite differences). We summarize the four main steps of the algorithm for a function $f(x)$, and we refer to [26] for additional details.

**Algorithm 2** Heat Geodesic Embedding

---

1: **Input:** $N \times d$ dataset matrix $\boldsymbol{X}$, denoising parameter $\rho \in [0,1]$, Harnack regularization $\sigma > 0$,
   output dimension $k$.
2: **Returns:** $N \times e$ embedding matrix $\boldsymbol{E}$.
3: ▷ *1. Calculate Heat Operator $\boldsymbol{H}_t$* ◁
4: **if** $t$ is "auto" **then**
5:     $t \leftarrow \text{Kneedle}\{H(\boldsymbol{H}_t)\}_t$          ▷ *Knee detection e.g. [26]*
6: $\boldsymbol{W} \leftarrow \text{kernel}(\boldsymbol{X})$
7: $\boldsymbol{L} \leftarrow \boldsymbol{Q} - \boldsymbol{W}$
8: **if** Exact **then**
9:     $\boldsymbol{H}_t \leftarrow \Psi e^{-t\Lambda} \Psi^T$
10: **else**
11:     $\boldsymbol{H}_t \leftarrow p_K(\boldsymbol{L}, t)$
12: ▷ *2. Calculate Pairwise Distances $\boldsymbol{D}$* ◁
13: $\boldsymbol{D} \leftarrow -4t \log \boldsymbol{H}_t$          ▷ $\log$ *is applied elementwise*
14: $\boldsymbol{D} \leftarrow (1-\rho)\boldsymbol{D} + \rho D_\text{T}$          ▷ *Triplet interpolation step*
15: Return $\boldsymbol{E} \leftarrow \text{MetricMDS}(\boldsymbol{D}, \|\cdot\|_2, k)$

---

1. Smoothing with a spline to preserve the shape of the function.

2. Normalize the values, so the algorithm does not depend on the magnitude of the observations.

3. Computing the set of finite differences for $x$ and $y := f(x)$, e.g. $y_{d_i} := f(x_i) - x_i$.

4. Evaluating local maxima of the difference curve $y_{d_i}$, and select the knee-point using a threshold based on the average difference between consecutive $x$.

## B    Experiments and datasets details

Our experiments compare our approach with multiple state-of-the-art baselines for synthetic datasets (for which the true geodesic distance is known) and real-world datasets. For all models, we perform sample splitting with a 50/50 validation-test split. The validation and test sets each consist of 5 repetitions with different random initializations. The hyper-parameters are selected according to the performance on the validation set. We always report the results on the test set, along with the standard deviations computed over the five repetitions. We use the following state-of-the-art methods in our experiments: our Heat Geodesic Embedding, *diffusion maps*[5], *PHATE* [22], *Heat-PHATE* (a variation of PHATE using the Heat Kernel), *Rand-Geo* (a variation of Heat Geodesic Embedding where we use the random walk kernel), *Shortest-path* which estimates the geodesic distance by computing the shortest path between two nodes in a graph built on the point clouds, *t-SNE*[32], and *UMAP*[19].

### B.1    Datasets

We consider two synthetic datasets, the well known Swiss roll and the tree datasets. The exact geodesic distance can be computed for these datasets. We additionally consider real-world datasets: PBMC, IPSC [22], EB [22], and two from the from the 2022 NeurIPS multimodal single-cell integration challenge[4].

### B.1.1    Swiss Roll

The Swiss roll dataset consists of data points samples on a smooth manifold inspired by shape of the famous alpine pastry. In its simplest form, it is a 2-dimensional surface embedded in $\mathbb{R}^3$ given by

$$
\begin{aligned}
x &= t \cdot cos(t) \\
y &= h \\
z &= t \cdot sin(t)
\end{aligned}
$$

---

[4]`https://www.kaggle.com/competitions/open-problems-multimodal/`

where $t \in [T_0, T_1]$ and $h \in [0, W]$. In our experiments we used $T_0 = \frac{3}{2}\pi$, $T_1 = \frac{9}{2}\pi$, and $W = 5$. We use two sampling mechanisms for generating the data points : uniformly and clustered. In the first, we sample points uniformly at random in the $[T_0, T_1] \times [0, W]$ plane. In the second, we sample according to a mixture of isotropic multivariate Gaussian distributions in the same plane with equal weights, means $[(7, W/2), (12, W/2)]$, and standard deviations $[1, 1]$. In the clustered case, data samples are given a label $y$ according to the Gaussian mixture component from which they were sampled.

We consider variations of the Swiss roll by projecting the data samples in higher dimension using a random rotation matrix sampled from the Haar distribution. We use three different ambient dimensions: 3, 10, and 50.

Finally, we add isotropic Gaussian noise to the data points in the ambient space with a standard deviation $\sigma$.

### B.1.2 Tree

The tree dataset is created by generating $K$ branches from a $D$-dimensional Brownian motion that are eventually glued together. Each branch is sampled from a multidimensional Brownian motion $d\mathbf{X_k} = 2d\mathbf{W}(t)$ at times $t = 0, 1, 2, ..., L - 1$ for $k \in [K]$. The first branch is taken as the main branch and the remaining branches are glued to the main branch by setting $X_k = X_k + X_0[i_k]$ where $i_k$ is a random index of the main branch vector. The total number of samples is thus $L \cdot K$

In our experiments, we used $L = 500$, $K = 5$, and $D = 5, 10$ (*i.e.,* two versions with different dimensions of the ambient space).

### B.2 Evaluation Metrics

We compare the performance of the different methods according to several metrics. For synthetic datasets, where ground truth geodesic distance is available, we directly compare the estimated distance matrices and ground truth geodesic distance matrices. For real-world datasets, we use clustering quality and continuous interpolation as evaluation metrics.

### B.2.1 Distance matrix evaluation

The following methods use an explicit distance matrix: diffusion maps, Heat Geodesic Embedding, Heat-Phate, Phate, Rand-Geo and Shortest Path. For these methods, we compare their ability their ability to recover the ground truth distance matrix several metrics. Letting $D$ and $\hat{D}$ the ground truth and inferred distance matrices respectively, and $N$ the number of points in the dataset, we use the following metrics.

**Pearson** $\rho$   We compute the average Pearson correlation between the rows of the distance matrices, $\frac{1}{N}\sum_{i=1}^{N} r_{D_i, \hat{D}_i}$, where $r_{x,y}$ is the Pearson correlation coefficient between vectors $x$ and $y$. $D_i$ stands for the $i$-th row of $D$.

**Spearman** $\rho$   We compute the average Spearman correlation between the rows of the distance matrices, $\frac{1}{N}\sum_{i=1}^{N} r_{D_i, \hat{D}_i}$, where $r_{x,y}$ is the Spearman correlation coefficient between vectors $x$ and $y$. $D_i$ stands for the $i$-th row of $D$.

**Frobenius Norm**   We use $\|D - \hat{D}\|_F$, where $\|A\|_F = \sqrt{\sum_{i=1}^{N}\sum_{j=1}^{N}|A_{i,j}|^2}$

**Maximum Norm**   We use $\|D - \hat{D}\|_\infty$, where $\|A\|_\infty = max_{i,j}|A_{i,j}|$

### B.2.2 Embedding evaluation

Some methods produce low-dimensional embeddings without using an explicit distance matrix for the data points. This is the case for UMAP and t-SNE. To compare against these methods, we use the distance matrix obtained by considering Euclidean distance between the low-dimensional embeddings. We used 2-dimensional embeddings in our experiments. For diffusion maps, we obtain

these embeddings by using the first two eigenvectors of the diffusion operator only. For Heat Geodesic Embedding, Heat-PHATE, PHATE, Rand-GEO and Shortest Path, we use multidimensional scaling (MDS) on the originally inferred distance matrix.

**Clustering**  We evaluate the ability of Heat Geodesic Embedding to create meaningful embeddings when clusters are present in the data. To this end, we run a k-means clustering on the two dimensional embeddings obtained with each method and compare them against the ground truth labels. For the Tree dataset, we use the branches as clusters. For the Swiss roll dataset, we sample data points on the manifold according to a mixture of Gaussians and use the mixture component as the ground truth cluster label.

**Interpolation**  To quantitatively evaluate the quality of the continuous embeddings, we first embed the entire dataset and obfuscate all samples from a particular time point (*e.g.*, $t = 2$). We then estimate the distribution of the missing time point by using displacement interpolation [35] between the adjacent time points (*e.g.*, $t = 1$ and $t = 3$). We report the Earth Mover Distance (EMD) between the predicted distribution and true distribution. A low EMD suggests that the obfuscated embeddings are naturally located between the previous and later time points, and that the generated embedding captures the temporal evolution of the data adequately.

### B.3  Hyperparameters

In Table 5, we report the values of hyperparameters used to compute the different embeddings.

| Hyperparameter | Description | Values |
|---|---|---|
| | Heat Geodesic Embedding | |
| k | Number of neighbours in k-NN graph | 5,10,15 |
| order | order of the approximation | 30 |
| $t$ | Diffusion time | 0.1,1,10,50,auto |
| Approximation method | Approximation method for Heat Kernel | Euler, Chebyshev |
| Laplacian | Type of laplacian | Combinatorial |
| Harnack $\rho$ | Harnack Regularization | 0,0.25,0.5,0.75,1,1.5 |
| | PHATE | |
| n-PCA | Number of PCA components | 50,100 |
| $t$ | Diffusion time | 1,5,10,20,auto |
| $k$ | Number of neighbours | 10 |
| | Diffusion Maps | |
| k | Number of neighbours in k-NN graph | 5,10,15 |
| $t$ | Diffusion time | 1,5,10,20 |
| | Shortest Path | |
| k | Number of neighbours in k-NN graph | 5,10,15 |
| | UMAP | |
| k | Number of neighbours | 5,10,15 |
| min-dist | Minimum distance | 0.1,0.5,0.99 |
| | t-SNE | |
| p | Perplexity | 10,30,100 |
| early exageration | Early exageration parameter | 12 |

Table 5: Hyperparameters used in our experiments

### B.4  Hardware

The experiments were performed on a compute node with 16 Intel Xeon Platinum 8358 Processors and 64GB RAM.

# C    Additional results

## C.1    HeatGeo weighted

Following Sec. 5, we know that weighting the MDS loss by the heat kernel corresponds to a specific parametrization of SNE, and thus promote the identification of cluster. In Fig. 5, we show the embeddings of four Gaussian distributions in 10 dimensions (top), and the PBMC dataset (bottom). The reference embedding is using t-SNE, as it models as it also minimizes the KL between the ambient and embedded distributions. We see that HeatGeo weighted form cluster that are shaped like a Gaussian. This is expected as Prop. 5.5, indicates that this is equivalent to minimizing the $D_{\mathrm{KL}}$ between the heat kernel and a Gaussian affinity kernel.

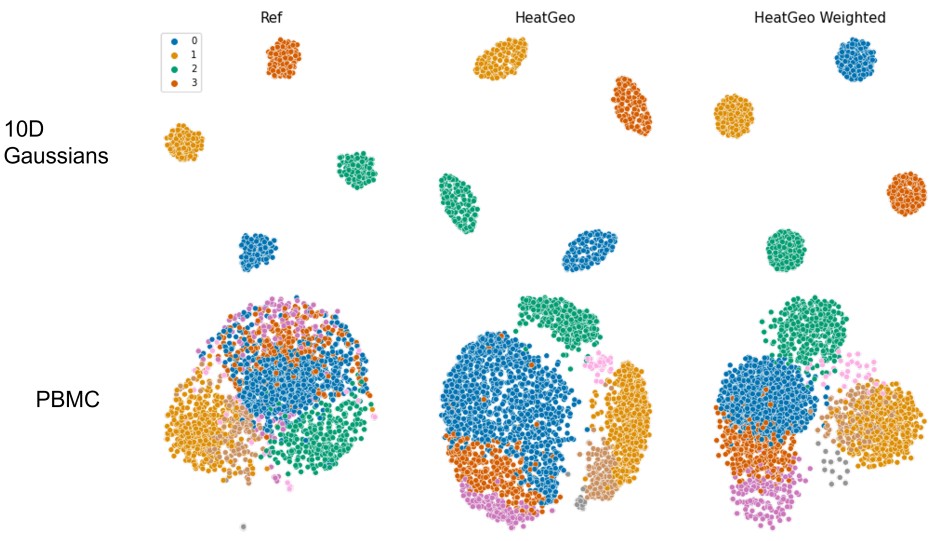

Figure 5: Embeddings of four Gaussian distributions in 10 dimensions (top), and the PBMC dataset (bottom). HeatGeo with weight is equivalent to minimizing the $D_{\mathrm{KL}}$ between the heat kernel and a Gaussian affinity kernel, hence produces clusters shaped similar to a Gaussian.

## C.2    Truncated distance

In Fig.6, we discretize the interval $[0, 51]$ in 51 nodes, and we compute the heat-geodesic distance of the midpoint with respect to the other points, effectively approximating the Euclidean distance. Using Chebyshev polynomials of degree of 20, we see that the impact of the truncation is greater as the diffusion time increases. The backward Euler methods does not result in a truncated distance.

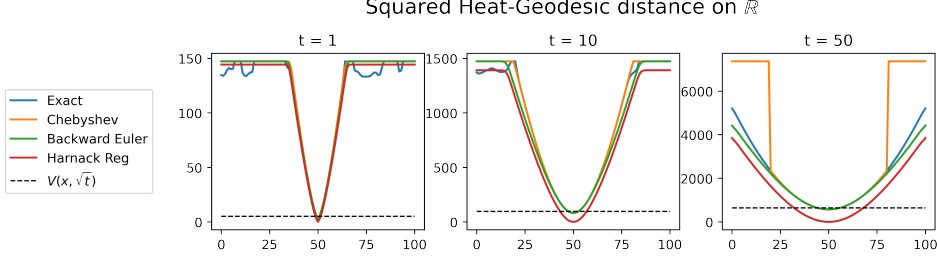

Figure 6: Approximation of the squared Euclidean distance with the Heat-geodesic for the exact computation, Backward Euler approximation, and Chebyshev polynomials. For larger diffusion time, the Chebyshev approximation results in a thresholded distance. The Harnack regularization unsures $d_t(x, x) = 0$.

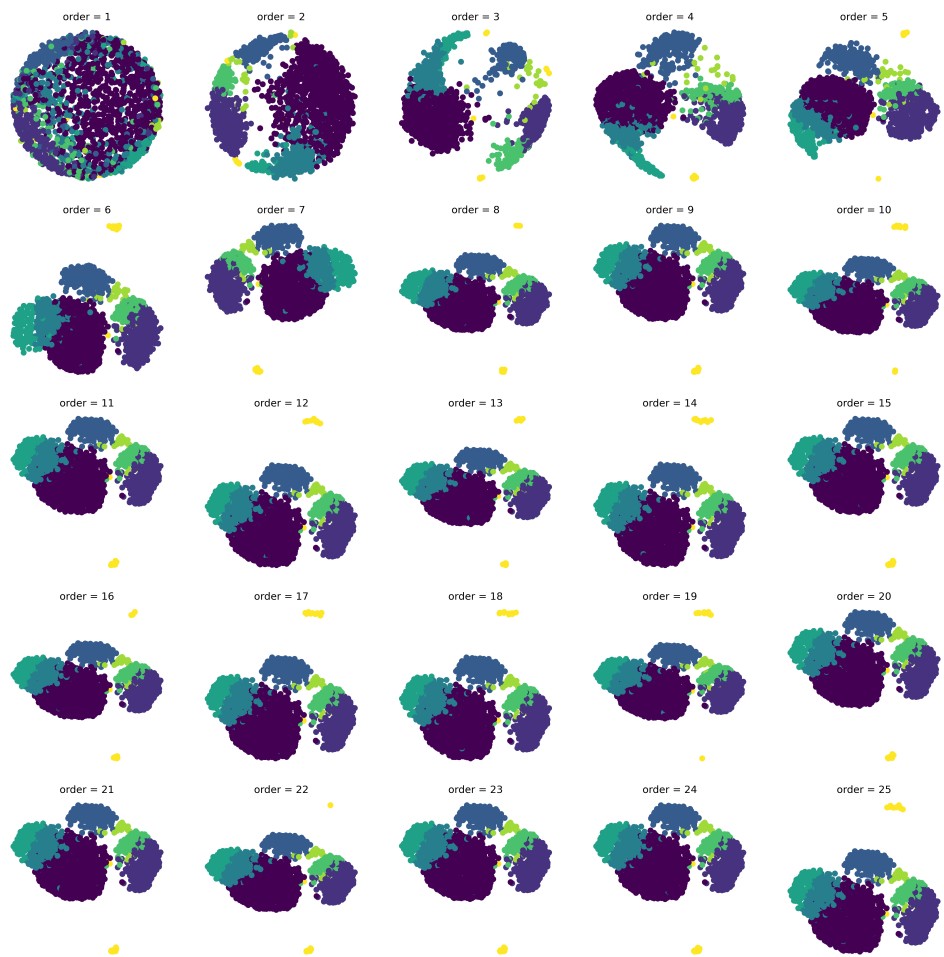

Figure 7: Impact of the Checbyshev approximation order on the embedding of HeatGeo for the PBMC dataset.

## C.3 Harnack inequality

For complete Riemannian manifolds that satisfy the parabolic Harnack inequality (PHI) we have $h_t(x, y) \simeq V^{-1}(x, \sqrt{t}) \, e^{-d(x,y)^2/t}$ so that $-t \log h_t(x, y) \simeq t \log V(x, \sqrt{t}) + d^2(x, y)$ [25].

$$h_t(x, x) = \frac{1}{V(x, \sqrt{t})} \tag{8}$$

$$V(x, \sqrt{t}) = h_t(x, x)^{-1} \tag{9}$$

We then have,

$$d^2(x, y) \simeq -t \log h_t(x, y) - t \log V(x, \sqrt{t})$$
$$d^2(x, y) \simeq -t \log h_t(x, y) - t \log h_t(x, x)^{-1}$$
$$d^2(x, y) \simeq -t \log h_t(x, y) + t \log h_t(x, x)$$

### C.3.1 Case studies for specific manifolds

**The circle -** $\mathbb{S}_1$    We now show that our expression for the Heat Geodesic Embedding-distance is monotonically increasing with respect to the ground truth geodesic distance $d \in \mathbb{R}^+$ for a fixed diffusion time $t$ and for any Harnack regularization in $\mathbb{S}_1$.

Our expression for the Heat Geodesic Embedding-distance is

$$\hat{d} = \sqrt{-4t \log(h_t(d)) + 4t \log(h_t(0))}$$

As the square-root is monotonic, and $4t \log h_t(0)$ is constant with respect to $d$, we need to show that $f(d) = -log(h_t(d))$ is monotonically increasing.

For $\mathbb{S}_1$, we have

$$h_t(d) = \sum_{m \in \mathbb{Z}} \frac{1}{\sqrt{4\pi t}} e^{-\frac{(d+2\pi m)^2}{4t}}$$

As log is monotonically increasing, it suffices to show that $\sum_{m \in \mathbb{Z}} e^{-\frac{(d+2\pi m)^2}{4t}}$ is monotonically *decreasing*, which is the case as for any $d' > d$, $\forall m \in \mathbb{Z}$, we have

$$e^{-\frac{(d+2\pi m)^2}{4t}} > e^{-\frac{(d'+2\pi m)^2}{4t}}.$$

In general, one can see that (1) the heat kernel depending only on the geodesic distance and (2) the heat kernel being monotonically decreasing with respect to the geodesic distance are sufficient conditions for preserving ordering of pair-wise distances with Heat Geodesic Embedding.

**The sphere - $\mathbb{S}_n$**  The above result can be applied to the higher-dimensional sphere $\mathbb{S}_n$. It is known that the heat kernel on manifold of constant curvatures is a function of the the geodesic distance ($d$) and time only. For $\mathbb{S}_n$ the heat kernel is given by

$$h_t(x, y) = \sum_{l=0}^{\infty} e^{-l(l+n)-2t} \frac{2l + n - 2}{n - 2} C_l^{\frac{n}{2}-1}(cos(d))$$

with $I$ the regularized incomplete beta function and $C$ the Gegenbauer polynomials.

Furthermore, Nowak et al. [24] showed that the heat kernel of the sphere is monotonically decreasing. The distance inferred from Heat Geodesic Embedding thus preserves ordering of the pair-wise distances.

**Euclidean ($\mathbb{R}^3$)**  For the euclidean space, we have for the volume of $\sqrt{t}$-geodesic ball and for the heat kernel:

$$V_{\sqrt{t}} = \frac{4}{3}\pi t^{3/2}$$
$$h_t(x, y) = \frac{1}{(4\pi t)^{3/2}} e^{-\frac{\rho^2}{4t}}.$$

Recalling Harnack inequality,

$$\frac{c_1}{V(x, \sqrt{t})} e^{-\frac{d(x,y)^2}{c_2 t}} \le h_t(x, y) \le \frac{c_3}{V(x, \sqrt{t})} e^{-\frac{d(x,y)^2}{c_4 t}}$$

With $c_2 = c_4 = 4$, we have

$$\frac{c_1}{V(x, \sqrt{t})} \leq \frac{1}{(4\pi t)^{3/2}} \leq \frac{c_3}{V(x, \sqrt{t})}$$

In this case, the bound can be made tight, by setting

$$\begin{aligned}
c_1 = c_3 &= \frac{V(x, \sqrt{t})}{(4\pi t)^{3/2}} \\
&= \frac{\frac{4}{3}\pi t^{3/2}}{(4\pi t)^{3/2}} \\
&= \frac{1}{3\sqrt{4\pi}} = \frac{1}{6\sqrt{\pi}},
\end{aligned}$$

we recover the exact geodesic distance.

### C.4  Quantitative results

#### C.4.1  Distance matrix evaluation

We report the performance of the different methods in terms of the ground truth geodesic matrix reconstruction in Table. 6 for the Swiss roll dataset and in Table. 7, for the Tree dataset.

#### C.4.2  Distance matrix evaluation via two-dimensional embeddings

We report the performance of the different methods in terms of the ground truth geodesic matrix reconstruction in Table 8 for the Swiss roll dataset and in Table 9, for the Tree dataset.

#### C.4.3  Clustering quality evaluation

On Tables 10, we report the performance on clustering quality for the synthetic datasets with different noise level.

### C.5  Impact of the different hyperparameters

We investigate the impact of the different hyperparameters on the quality of the embeddings. In Figure 8, we show the embeddings of HeatGeo for different values of diffusion time, number of neighbours, order, and Harnack regularization.

In Figures 9, 10, 11, and 12, we show the impact of different hyperparameters on the Pearson correlation between the estimated distance matrix and ground truth distance matrix for different methods on the Swiss roll dataset.

### C.6  Graph construction

We compare the embeddings of the heat-geodesic distance for different graph construction. Throughout the paper we used the graph construction from PHATE [22]. In the following we present additional results depending on the choice of kernel to construct the graph. Specifically, we use a simple nearest neighbor (kNN) graph implemented in [7], the graph from UMAP [19], and the implementation in the package Scanpy [36] for single-cell analysis. In figure, we present the embeddings 2500 points of a tree with five branches in 10 dimensions, where the observations are perturbed with a standard Gaussian noise. All methods used five nearest neighbors and a diffusion time of 20. In Figure 13, we show the evolution of the Pearson correlation between estimated and ground truth distance matrices for the 10-dimensional Swiss roll dataset for various graph constructions. We note that the results are stable across different graph construction strategies.

| data | Noise level | Method | PearsonR | SpearmanR | Norm Fro N2 | Norm inf N2 |
|---|---|---|---|---|---|---|
| Swiss roll | 0.1 | Diffusion Map | $0.974 \pm 0.01$ | $0.983 \pm 0.007$ | $0.018 \pm 0.0$ | $0.026 \pm 0.0$ |
| Swiss roll | 0.1 | Heat-Geo | $0.992 \pm 0.003$ | $0.995 \pm 0.002$ | $0.002 \pm 0.0$ | $0.003 \pm 0.0$ |
| Swiss roll | 0.1 | Heat-PHATE | $0.99 \pm 0.002$ | $0.997 \pm 0.001$ | $0.079 \pm 0.002$ | $0.1 \pm 0.003$ |
| Swiss roll | 0.1 | PHATE | $0.621 \pm 0.006$ | $0.58 \pm 0.01$ | $0.022 \pm 0.0$ | $0.026 \pm 0.0$ |
| Swiss roll | 0.1 | Rand-Geo | $0.956 \pm 0.003$ | $0.993 \pm 0.001$ | $0.009 \pm 0.0$ | $0.012 \pm 0.0$ |
| Swiss roll | 0.1 | Shortest Path | $\mathbf{1.0 \pm 0.0}$ | $\mathbf{1.0 \pm 0.0}$ | $\mathbf{0.0 \pm 0.0}$ | $\mathbf{0.001 \pm 0.0}$ |
| Swiss roll | 0.1 | Euclidean | $0.379 \pm 0.003$ | $0.424 \pm 0.003$ | $0.014 \pm 0.0$ | $0.018 \pm 0.0$ |
| Swiss roll | 0.5 | Diffusion Map | $0.982 \pm 0.003$ | $0.987 \pm 0.002$ | $0.018 \pm 0.0$ | $0.026 \pm 0.0$ |
| Swiss roll | 0.5 | Heat-Geo | $0.994 \pm 0.002$ | $0.996 \pm 0.001$ | $0.002 \pm 0.0$ | $0.004 \pm 0.0$ |
| Swiss roll | 0.5 | Heat-PHATE | $0.993 \pm 0.001$ | $0.998 \pm 0.0$ | $0.064 \pm 0.001$ | $0.083 \pm 0.002$ |
| Swiss roll | 0.5 | PHATE | $0.649 \pm 0.007$ | $0.615 \pm 0.006$ | $0.023 \pm 0.0$ | $0.028 \pm 0.0$ |
| Swiss roll | 0.5 | Rand-Geo | $0.969 \pm 0.002$ | $0.995 \pm 0.001$ | $0.009 \pm 0.0$ | $0.011 \pm 0.0$ |
| Swiss roll | 0.5 | Shortest Path | $\mathbf{0.999 \pm 0.0}$ | $\mathbf{0.999 \pm 0.0}$ | $\mathbf{0.001 \pm 0.0}$ | $\mathbf{0.002 \pm 0.0}$ |
| Swiss roll | 0.5 | Euclidean | $0.376 \pm 0.004$ | $0.422 \pm 0.004$ | $0.013 \pm 0.0$ | $0.018 \pm 0.0$ |
| Swiss roll | 1.0 | Diffusion Map | $0.476 \pm 0.226$ | $0.478 \pm 0.138$ | $0.018 \pm 0.0$ | $0.026 \pm 0.0$ |
| Swiss roll | 1.0 | Heat-Geo | $\mathbf{0.702 \pm 0.086}$ | $\mathbf{0.7 \pm 0.073}$ | $\mathbf{0.01 \pm 0.0}$ | $\mathbf{0.012 \pm 0.0}$ |
| Swiss roll | 1.0 | Heat-PHATE | $0.623 \pm 0.144$ | $0.633 \pm 0.114$ | $\mathbf{0.01 \pm 0.002}$ | $0.019 \pm 0.004$ |
| Swiss roll | 1.0 | PHATE | $0.457 \pm 0.01$ | $0.404 \pm 0.024$ | $0.024 \pm 0.0$ | $0.028 \pm 0.0$ |
| Swiss roll | 1.0 | Rand-Geo | $0.521 \pm 0.042$ | $0.608 \pm 0.025$ | $\mathbf{0.01 \pm 0.0}$ | $0.014 \pm 0.0$ |
| Swiss roll | 1.0 | Shortest Path | $0.497 \pm 0.144$ | $0.558 \pm 0.134$ | $0.011 \pm 0.001$ | $0.015 \pm 0.002$ |
| Swiss roll | 1.0 | Euclidean | $0.365 \pm 0.006$ | $0.413 \pm 0.005$ | $0.013 \pm 0.0$ | $0.019 \pm 0.001$ |
| Swiss roll high | 0.1 | Diffusion Map | $0.98 \pm 0.003$ | $0.986 \pm 0.001$ | $0.018 \pm 0.0$ | $0.026 \pm 0.0$ |
| Swiss roll high | 0.1 | Heat-Geo | $0.992 \pm 0.003$ | $0.996 \pm 0.002$ | $0.002 \pm 0.0$ | $0.003 \pm 0.0$ |
| Swiss roll high | 0.1 | Heat-PHATE | $0.991 \pm 0.002$ | $0.997 \pm 0.001$ | $0.079 \pm 0.002$ | $0.101 \pm 0.004$ |
| Swiss roll high | 0.1 | PHATE | $0.625 \pm 0.013$ | $0.582 \pm 0.017$ | $0.022 \pm 0.0$ | $0.026 \pm 0.0$ |
| Swiss roll high | 0.1 | Rand-Geo | $0.956 \pm 0.002$ | $0.993 \pm 0.001$ | $0.009 \pm 0.0$ | $0.012 \pm 0.0$ |
| Swiss roll high | 0.1 | Shortest Path | $\mathbf{1.0 \pm 0.0}$ | $\mathbf{1.0 \pm 0.0}$ | $\mathbf{0.001 \pm 0.0}$ | $\mathbf{0.002 \pm 0.0}$ |
| Swiss roll high | 0.1 | Euclidean | $0.379 \pm 0.002$ | $0.424 \pm 0.002$ | $0.014 \pm 0.0$ | $0.018 \pm 0.0$ |
| Swiss roll high | 0.5 | Diffusion Map | $0.98 \pm 0.002$ | $0.985 \pm 0.002$ | $0.018 \pm 0.0$ | $0.026 \pm 0.0$ |
| Swiss roll high | 0.5 | Heat-Geo | $0.997 \pm 0.001$ | $0.997 \pm 0.0$ | $\mathbf{0.005 \pm 0.0}$ | $\mathbf{0.007 \pm 0.0}$ |
| Swiss roll high | 0.5 | Heat-PHATE | $0.995 \pm 0.0$ | $0.997 \pm 0.0$ | $0.041 \pm 0.001$ | $0.054 \pm 0.002$ |
| Swiss roll high | 0.5 | PHATE | $0.717 \pm 0.004$ | $0.707 \pm 0.005$ | $0.026 \pm 0.0$ | $0.034 \pm 0.001$ |
| Swiss roll high | 0.5 | Rand-Geo | $0.984 \pm 0.0$ | $0.996 \pm 0.0$ | $0.008 \pm 0.0$ | $0.01 \pm 0.0$ |
| Swiss roll high | 0.5 | Shortest Path | $\mathbf{0.999 \pm 0.0}$ | $\mathbf{0.998 \pm 0.0}$ | $0.006 \pm 0.0$ | $0.009 \pm 0.0$ |
| Swiss roll high | 0.5 | Euclidean | $0.369 \pm 0.003$ | $0.421 \pm 0.003$ | $0.013 \pm 0.0$ | $0.018 \pm 0.0$ |
| Swiss roll high | 1.0 | Diffusion Map | $0.555 \pm 0.155$ | $0.526 \pm 0.081$ | $0.018 \pm 0.0$ | $0.026 \pm 0.0$ |
| Swiss roll high | 1.0 | Heat-Geo | $\mathbf{0.705 \pm 0.065}$ | $\mathbf{0.695 \pm 0.052}$ | $0.011 \pm 0.0$ | $\mathbf{0.012 \pm 0.0}$ |
| Swiss roll high | 1.0 | Heat-PHATE | $0.63 \pm 0.106$ | $0.625 \pm 0.074$ | $0.011 \pm 0.001$ | $0.014 \pm 0.002$ |
| Swiss roll high | 1.0 | PHATE | $0.473 \pm 0.026$ | $0.419 \pm 0.024$ | $0.027 \pm 0.0$ | $0.039 \pm 0.001$ |
| Swiss roll high | 1.0 | Rand-Geo | $0.563 \pm 0.05$ | $0.644 \pm 0.033$ | $\mathbf{0.01 \pm 0.0}$ | $\mathbf{0.012 \pm 0.0}$ |
| Swiss roll high | 1.0 | Shortest Path | $0.384 \pm 0.02$ | $0.461 \pm 0.017$ | $0.011 \pm 0.0$ | $0.015 \pm 0.0$ |
| Swiss roll high | 1.0 | Euclidean | $0.349 \pm 0.004$ | $0.409 \pm 0.003$ | $0.013 \pm 0.0$ | $0.018 \pm 0.0$ |
| Swiss roll very high | 0.1 | Diffusion Map | $0.977 \pm 0.005$ | $0.984 \pm 0.004$ | $0.018 \pm 0.0$ | $0.026 \pm 0.0$ |
| Swiss roll very high | 0.1 | Heat-Geo | $0.992 \pm 0.002$ | $0.996 \pm 0.001$ | $\mathbf{0.002 \pm 0.0}$ | $\mathbf{0.003 \pm 0.0}$ |
| Swiss roll very high | 0.1 | Heat-PHATE | $0.991 \pm 0.001$ | $0.997 \pm 0.001$ | $0.079 \pm 0.003$ | $0.101 \pm 0.003$ |
| Swiss roll very high | 0.1 | PHATE | $0.631 \pm 0.01$ | $0.594 \pm 0.011$ | $0.023 \pm 0.0$ | $0.028 \pm 0.001$ |
| Swiss roll very high | 0.1 | Rand-Geo | $0.957 \pm 0.002$ | $0.994 \pm 0.001$ | $0.009 \pm 0.0$ | $0.012 \pm 0.0$ |
| Swiss roll very high | 0.1 | Shortest Path | $\mathbf{0.999 \pm 0.0}$ | $\mathbf{0.999 \pm 0.0}$ | $0.006 \pm 0.0$ | $0.007 \pm 0.0$ |
| Swiss roll very high | 0.1 | Euclidean | $0.378 \pm 0.002$ | $0.424 \pm 0.002$ | $0.013 \pm 0.0$ | $0.018 \pm 0.0$ |
| Swiss roll very high | 0.5 | Diffusion Map | $0.978 \pm 0.002$ | $0.984 \pm 0.001$ | $0.018 \pm 0.0$ | $0.026 \pm 0.0$ |
| Swiss roll very high | 0.5 | Heat-Geo | $0.997 \pm 0.0$ | $\mathbf{0.998 \pm 0.0}$ | $0.008 \pm 0.0$ | $0.01 \pm 0.0$ |
| Swiss roll very high | 0.5 | Heat-PHATE | $0.996 \pm 0.001$ | $0.997 \pm 0.0$ | $0.016 \pm 0.0$ | $0.02 \pm 0.001$ |
| Swiss roll very high | 0.5 | PHATE | $0.815 \pm 0.002$ | $0.823 \pm 0.004$ | $0.032 \pm 0.0$ | $0.049 \pm 0.002$ |
| Swiss roll very high | 0.5 | Rand-Geo | $0.986 \pm 0.0$ | $0.996 \pm 0.0$ | $\mathbf{0.008 \pm 0.0}$ | $\mathbf{0.009 \pm 0.0}$ |
| Swiss roll very high | 0.5 | Shortest Path | $\mathbf{0.998 \pm 0.0}$ | $\mathbf{0.998 \pm 0.0}$ | $0.019 \pm 0.001$ | $0.027 \pm 0.001$ |
| Swiss roll very high | 0.5 | Euclidean | $0.361 \pm 0.002$ | $0.42 \pm 0.002$ | $0.013 \pm 0.0$ | $0.018 \pm 0.0$ |
| Swiss roll very high | 1.0 | Diffusion Map | $0.324 \pm 0.061$ | $0.399 \pm 0.033$ | $0.018 \pm 0.0$ | $0.026 \pm 0.0$ |
| Swiss roll very high | 1.0 | Heat-Geo | $\mathbf{0.466 \pm 0.007}$ | $0.506 \pm 0.006$ | $0.011 \pm 0.0$ | $0.013 \pm 0.0$ |
| Swiss roll very high | 1.0 | Heat-PHATE | $0.369 \pm 0.011$ | $0.43 \pm 0.019$ | $0.011 \pm 0.0$ | $0.014 \pm 0.0$ |
| Swiss roll very high | 1.0 | PHATE | $0.377 \pm 0.011$ | $0.425 \pm 0.009$ | $0.036 \pm 0.0$ | $0.062 \pm 0.004$ |
| Swiss roll very high | 1.0 | Rand-Geo | $0.398 \pm 0.009$ | $\mathbf{0.516 \pm 0.008}$ | $\mathbf{0.01 \pm 0.0}$ | $\mathbf{0.012 \pm 0.0}$ |
| Swiss roll very high | 1.0 | Shortest Path | $0.367 \pm 0.018$ | $0.443 \pm 0.016$ | $0.012 \pm 0.0$ | $0.015 \pm 0.0$ |
| Swiss roll very high | 1.0 | Euclidean | $0.336 \pm 0.002$ | $0.402 \pm 0.002$ | $0.012 \pm 0.0$ | $0.018 \pm 0.0$ |

Table 6: Comparison of the estimated distance matrices with the ground truth geodesic distance matrices on the Swiss roll dataset. Best models on average are bolded (not necessarily significant).

| data | Noise level | Method | PearsonR | SpearmanR | Norm Fro N2 | Norm inf N2 |
|---|---|---|---|---|---|---|
| Tree | 1.0 | Diffusion Map | $0.748 \pm 0.125$ | $0.733 \pm 0.111$ | $0.113 \pm 0.012$ | $0.161 \pm 0.019$ |
| Tree | 1.0 | Heat-Geo | $\mathbf{0.976 \pm 0.019}$ | $\mathbf{0.977 \pm 0.02}$ | $0.092 \pm 0.011$ | $0.135 \pm 0.018$ |
| Tree | 1.0 | Heat-PHATE | $0.918 \pm 0.032$ | $0.885 \pm 0.04$ | $\mathbf{0.03 \pm 0.005}$ | $\mathbf{0.044 \pm 0.007}$ |
| Tree | 1.0 | PHATE | $0.671 \pm 0.021$ | $0.398 \pm 0.052$ | $0.051 \pm 0.008$ | $0.084 \pm 0.017$ |
| Tree | 1.0 | Rand-Geo | $0.926 \pm 0.011$ | $0.966 \pm 0.019$ | $0.076 \pm 0.01$ | $0.117 \pm 0.018$ |
| Tree | 1.0 | Shortest Path | $0.965 \pm 0.026$ | $0.963 \pm 0.027$ | $0.039 \pm 0.008$ | $0.06 \pm 0.008$ |
| Tree | 1.0 | Euclidean | $0.508 \pm 0.039$ | $0.483 \pm 0.052$ | $0.092 \pm 0.011$ | $0.138 \pm 0.018$ |
| Tree | 5.0 | Diffusion Map | $0.656 \pm 0.054$ | $0.653 \pm 0.057$ | $0.113 \pm 0.012$ | $0.161 \pm 0.019$ |
| Tree | 5.0 | Heat-Geo | $\mathbf{0.822 \pm 0.008}$ | $\mathbf{0.807 \pm 0.016}$ | $0.1 \pm 0.012$ | $0.146 \pm 0.019$ |
| Tree | 5.0 | Heat-PHATE | $0.765 \pm 0.025$ | $0.751 \pm 0.023$ | $\mathbf{0.043 \pm 0.006}$ | $\mathbf{0.08 \pm 0.01}$ |
| Tree | 5.0 | PHATE | $0.766 \pm 0.023$ | $0.743 \pm 0.028$ | $0.055 \pm 0.007$ | $0.093 \pm 0.008$ |
| Tree | 5.0 | Rand-Geo | $0.806 \pm 0.014$ | $0.795 \pm 0.018$ | $0.094 \pm 0.011$ | $0.139 \pm 0.018$ |
| Tree | 5.0 | Shortest Path | $0.78 \pm 0.009$ | $0.757 \pm 0.019$ | $0.075 \pm 0.009$ | $0.117 \pm 0.014$ |
| Tree | 5.0 | Euclidean | $0.735 \pm 0.014$ | $0.704 \pm 0.033$ | $0.096 \pm 0.011$ | $0.141 \pm 0.017$ |
| Tree | 10.0 | Diffusion Map | $0.538 \pm 0.05$ | $0.471 \pm 0.089$ | $0.113 \pm 0.012$ | $0.161 \pm 0.019$ |
| Tree | 10.0 | Heat-Geo | $0.62 \pm 0.025$ | $\mathbf{0.59 \pm 0.033}$ | $0.1 \pm 0.012$ | $0.146 \pm 0.019$ |
| Tree | 10.0 | Heat-PHATE | $\mathbf{0.63 \pm 0.018}$ | $0.588 \pm 0.031$ | $\mathbf{0.046 \pm 0.005}$ | $\mathbf{0.083 \pm 0.012}$ |
| Tree | 10.0 | PHATE | $0.623 \pm 0.016$ | $0.583 \pm 0.029$ | $0.07 \pm 0.01$ | $0.112 \pm 0.017$ |
| Tree | 10.0 | Rand-Geo | $0.578 \pm 0.043$ | $0.558 \pm 0.053$ | $0.095 \pm 0.011$ | $0.14 \pm 0.018$ |
| Tree | 10.0 | Shortest Path | $0.539 \pm 0.041$ | $0.513 \pm 0.055$ | $0.072 \pm 0.01$ | $0.118 \pm 0.017$ |
| Tree | 10.0 | Euclidean | $0.508 \pm 0.039$ | $0.483 \pm 0.052$ | $0.092 \pm 0.011$ | $0.138 \pm 0.018$ |
| Tree high | 1.0 | Diffusion Map | $0.754 \pm 0.049$ | $0.741 \pm 0.057$ | $0.267 \pm 0.021$ | $0.369 \pm 0.026$ |
| Tree high | 1.0 | Heat-Geo | $0.996 \pm 0.001$ | $\mathbf{0.999 \pm 0.001}$ | $0.242 \pm 0.02$ | $0.338 \pm 0.026$ |
| Tree high | 1.0 | Heat-PHATE | $0.927 \pm 0.011$ | $0.875 \pm 0.032$ | $0.062 \pm 0.003$ | $0.084 \pm 0.006$ |
| Tree high | 1.0 | PHATE | $0.528 \pm 0.085$ | $0.141 \pm 0.061$ | $0.209 \pm 0.023$ | $0.307 \pm 0.027$ |
| Tree high | 1.0 | Rand-Geo | $0.85 \pm 0.014$ | $0.944 \pm 0.011$ | $0.227 \pm 0.02$ | $0.323 \pm 0.025$ |
| Tree high | 1.0 | Shortest Path | $\mathbf{0.998 \pm 0.001}$ | $\mathbf{0.999 \pm 0.001}$ | $\mathbf{0.009 \pm 0.002}$ | $\mathbf{0.018 \pm 0.005}$ |
| Tree high | 1.0 | Euclidean | $0.928 \pm 0.018$ | $0.928 \pm 0.024$ | $0.24 \pm 0.02$ | $0.334 \pm 0.026$ |
| Tree high | 5.0 | Diffusion Map | $0.706 \pm 0.124$ | $0.705 \pm 0.113$ | $0.267 \pm 0.021$ | $0.369 \pm 0.026$ |
| Tree high | 5.0 | Heat-Geo | $\mathbf{0.97 \pm 0.01}$ | $\mathbf{0.975 \pm 0.009}$ | $0.253 \pm 0.021$ | $0.353 \pm 0.026$ |
| Tree high | 5.0 | Heat-PHATE | $0.932 \pm 0.022$ | $0.919 \pm 0.03$ | $\mathbf{0.072 \pm 0.004}$ | $\mathbf{0.112 \pm 0.008}$ |
| Tree high | 5.0 | PHATE | $0.913 \pm 0.014$ | $0.872 \pm 0.034$ | $0.19 \pm 0.017$ | $0.278 \pm 0.025$ |
| Tree high | 5.0 | Rand-Geo | $0.968 \pm 0.01$ | $0.971 \pm 0.009$ | $0.245 \pm 0.019$ | $0.342 \pm 0.024$ |
| Tree high | 5.0 | Shortest Path | $0.952 \pm 0.016$ | $0.95 \pm 0.019$ | $0.137 \pm 0.017$ | $0.209 \pm 0.024$ |
| Tree high | 5.0 | Euclidean | $0.882 \pm 0.028$ | $0.873 \pm 0.032$ | $0.237 \pm 0.02$ | $0.333 \pm 0.025$ |
| Tree high | 10.0 | Diffusion Map | $0.598 \pm 0.117$ | $0.613 \pm 0.103$ | $0.267 \pm 0.021$ | $0.369 \pm 0.026$ |
| Tree high | 10.0 | Heat-Geo | $\mathbf{0.861 \pm 0.039}$ | $\mathbf{0.87 \pm 0.038}$ | $0.254 \pm 0.021$ | $0.353 \pm 0.026$ |
| Tree high | 10.0 | Heat-PHATE | $0.844 \pm 0.05$ | $0.838 \pm 0.051$ | $0.168 \pm 0.015$ | $0.27 \pm 0.025$ |
| Tree high | 10.0 | PHATE | $0.837 \pm 0.052$ | $0.838 \pm 0.049$ | $0.204 \pm 0.018$ | $0.301 \pm 0.024$ |
| Tree high | 10.0 | Rand-Geo | $0.845 \pm 0.041$ | $0.86 \pm 0.038$ | $0.248 \pm 0.02$ | $0.346 \pm 0.025$ |
| Tree high | 10.0 | Shortest Path | $0.779 \pm 0.051$ | $0.777 \pm 0.054$ | $\mathbf{0.159 \pm 0.018}$ | $\mathbf{0.257 \pm 0.026}$ |
| Tree high | 10.0 | Euclidean | $0.709 \pm 0.054$ | $0.699 \pm 0.059$ | $0.229 \pm 0.02$ | $0.327 \pm 0.026$ |

Table 7: Comparison of the estimated distance matrices with the ground truth geodesic distance matrices on the Tree roll dataset. Best models on average are bolded (not necessarily significant).

## C.7 Time Complexity

In Table 12, we present the average computing time for creating embeddings and corresponding distance matrix for the different methods. All methods are applied on the Swiss roll dataset in three dimension with 2000 samples. We present empirical averages and standard deviations over ten repetitions. The experiments were run on a Apple M2 Pro chip with 16G RAM.

| data | Noise level | Method | PearsonR | SpearmanR | Norm Fro N2 | Norm inf N2 |
|---|---|---|---|---|---|---|
| Swiss roll | 0.1 | Diffusion Map | $0.974 \pm 0.01$ | $0.983 \pm 0.007$ | $0.018 \pm 0.0$ | $0.026 \pm 0.0$ |
| Swiss roll | 0.1 | Heat-Geo | $0.995 \pm 0.003$ | $0.996 \pm 0.002$ | $0.018 \pm 0.0$ | $0.026 \pm 0.0$ |
| Swiss roll | 0.1 | Heat-PHATE | $0.99 \pm 0.002$ | $0.997 \pm 0.001$ | $0.018 \pm 0.0$ | $0.026 \pm 0.0$ |
| Swiss roll | 0.1 | PHATE | $0.677 \pm 0.02$ | $0.697 \pm 0.014$ | $0.018 \pm 0.0$ | $0.026 \pm 0.0$ |
| Swiss roll | 0.1 | Rand-Geo | $0.917 \pm 0.003$ | $0.915 \pm 0.002$ | $0.018 \pm 0.0$ | $0.026 \pm 0.0$ |
| Swiss roll | 0.1 | Shortest Path | $\mathbf{1.0 \pm 0.0}$ | $\mathbf{1.0 \pm 0.0}$ | $0.018 \pm 0.0$ | $0.026 \pm 0.0$ |
| Swiss roll | 0.1 | TSNE | $0.905 \pm 0.005$ | $0.897 \pm 0.004$ | $\mathbf{0.006 \pm 0.0}$ | $\mathbf{0.008 \pm 0.0}$ |
| Swiss roll | 0.1 | UMAP | $0.802 \pm 0.013$ | $0.79 \pm 0.012$ | $0.011 \pm 0.0$ | $0.016 \pm 0.001$ |
| Swiss roll | 0.1 | Euclidean | $0.384 \pm 0.003$ | $0.424 \pm 0.003$ | $0.018 \pm 0.0$ | $0.026 \pm 0.0$ |
| Swiss roll | 0.5 | Diffusion Map | $0.982 \pm 0.003$ | $0.987 \pm 0.002$ | $0.018 \pm 0.0$ | $0.026 \pm 0.0$ |
| Swiss roll | 0.5 | Heat-Geo | $0.997 \pm 0.0$ | $0.996 \pm 0.001$ | $0.018 \pm 0.0$ | $0.026 \pm 0.0$ |
| Swiss roll | 0.5 | Heat-PHATE | $0.993 \pm 0.001$ | $0.997 \pm 0.0$ | $0.018 \pm 0.0$ | $0.026 \pm 0.0$ |
| Swiss roll | 0.5 | PHATE | $0.696 \pm 0.011$ | $0.711 \pm 0.008$ | $0.018 \pm 0.0$ | $0.026 \pm 0.0$ |
| Swiss roll | 0.5 | Rand-Geo | $0.932 \pm 0.002$ | $0.932 \pm 0.002$ | $0.018 \pm 0.0$ | $0.026 \pm 0.0$ |
| Swiss roll | 0.5 | Shortest Path | $\mathbf{0.999 \pm 0.0}$ | $\mathbf{0.999 \pm 0.0}$ | $0.018 \pm 0.0$ | $0.026 \pm 0.0$ |
| Swiss roll | 0.5 | TSNE | $0.899 \pm 0.01$ | $0.892 \pm 0.008$ | $\mathbf{0.006 \pm 0.0}$ | $\mathbf{0.008 \pm 0.0}$ |
| Swiss roll | 0.5 | UMAP | $0.838 \pm 0.019$ | $0.819 \pm 0.017$ | $0.012 \pm 0.0$ | $0.016 \pm 0.001$ |
| Swiss roll | 0.5 | Euclidean | $0.381 \pm 0.004$ | $0.421 \pm 0.004$ | $0.018 \pm 0.0$ | $0.026 \pm 0.0$ |
| Swiss roll | 1.0 | Diffusion Map | $0.476 \pm 0.226$ | $0.478 \pm 0.138$ | $0.018 \pm 0.0$ | $0.026 \pm 0.0$ |
| Swiss roll | 1.0 | Heat-Geo | $0.672 \pm 0.221$ | $0.676 \pm 0.193$ | $0.018 \pm 0.0$ | $0.026 \pm 0.0$ |
| Swiss roll | 1.0 | Heat-PHATE | $0.674 \pm 0.169$ | $0.684 \pm 0.134$ | $0.018 \pm 0.0$ | $0.026 \pm 0.0$ |
| Swiss roll | 1.0 | PHATE | $0.287 \pm 0.03$ | $0.349 \pm 0.028$ | $0.018 \pm 0.0$ | $0.026 \pm 0.0$ |
| Swiss roll | 1.0 | Rand-Geo | $0.39 \pm 0.029$ | $0.43 \pm 0.022$ | $0.018 \pm 0.0$ | $0.026 \pm 0.0$ |
| Swiss roll | 1.0 | Shortest Path | $0.467 \pm 0.17$ | $0.511 \pm 0.163$ | $0.018 \pm 0.0$ | $0.026 \pm 0.0$ |
| Swiss roll | 1.0 | TSNE | $0.721 \pm 0.183$ | $\mathbf{0.724 \pm 0.151}$ | $\mathbf{0.008 \pm 0.002}$ | $\mathbf{0.014 \pm 0.003}$ |
| Swiss roll | 1.0 | UMAP | $\mathbf{0.727 \pm 0.181}$ | $0.713 \pm 0.167$ | $0.012 \pm 0.001$ | $0.018 \pm 0.001$ |
| Swiss roll | 1.0 | Euclidean | $0.371 \pm 0.006$ | $0.414 \pm 0.005$ | $0.018 \pm 0.0$ | $0.026 \pm 0.0$ |
| Swiss roll | 5.0 | Diffusion Map | $0.157 \pm 0.021$ | $0.173 \pm 0.015$ | $0.018 \pm 0.0$ | $0.026 \pm 0.0$ |
| Swiss roll | 5.0 | Heat-PHATE | $0.203 \pm 0.014$ | $\mathbf{0.239 \pm 0.013}$ | $0.018 \pm 0.0$ | $0.026 \pm 0.0$ |
| Swiss roll | 5.0 | PHATE | $0.201 \pm 0.014$ | $0.237 \pm 0.013$ | $0.018 \pm 0.0$ | $0.026 \pm 0.0$ |
| Swiss roll | 5.0 | Rand-Geo | $0.201 \pm 0.014$ | $0.238 \pm 0.012$ | $0.018 \pm 0.0$ | $0.026 \pm 0.0$ |
| Swiss roll | 5.0 | Shortest Path | $0.2 \pm 0.011$ | $0.233 \pm 0.01$ | $0.018 \pm 0.0$ | $0.026 \pm 0.0$ |
| Swiss roll | 5.0 | TSNE | $0.2 \pm 0.011$ | $0.233 \pm 0.01$ | $\mathbf{0.012 \pm 0.0}$ | $\mathbf{0.018 \pm 0.001}$ |
| Swiss roll | 5.0 | UMAP | $\mathbf{0.205 \pm 0.013}$ | $\mathbf{0.239 \pm 0.012}$ | $0.015 \pm 0.0$ | $0.022 \pm 0.0$ |
| Swiss roll high | 0.1 | Diffusion Map | $0.98 \pm 0.003$ | $0.986 \pm 0.001$ | $0.018 \pm 0.0$ | $0.026 \pm 0.0$ |
| Swiss roll high | 0.1 | Heat-Geo | $0.996 \pm 0.002$ | $0.997 \pm 0.001$ | $0.018 \pm 0.0$ | $0.026 \pm 0.0$ |
| Swiss roll high | 0.1 | Heat-PHATE | $0.991 \pm 0.002$ | $0.997 \pm 0.001$ | $0.018 \pm 0.0$ | $0.026 \pm 0.0$ |
| Swiss roll high | 0.1 | PHATE | $0.678 \pm 0.027$ | $0.698 \pm 0.019$ | $0.018 \pm 0.0$ | $0.026 \pm 0.0$ |
| Swiss roll high | 0.1 | Rand-Geo | $0.917 \pm 0.003$ | $0.915 \pm 0.002$ | $0.018 \pm 0.0$ | $0.026 \pm 0.0$ |
| Swiss roll high | 0.1 | Shortest Path | $\mathbf{1.0 \pm 0.0}$ | $\mathbf{1.0 \pm 0.0}$ | $0.018 \pm 0.0$ | $0.026 \pm 0.0$ |
| Swiss roll high | 0.1 | TSNE | $0.903 \pm 0.004$ | $0.896 \pm 0.003$ | $\mathbf{0.006 \pm 0.0}$ | $\mathbf{0.008 \pm 0.0}$ |
| Swiss roll high | 0.1 | UMAP | $0.806 \pm 0.014$ | $0.794 \pm 0.01$ | $0.011 \pm 0.0$ | $0.016 \pm 0.001$ |
| Swiss roll high | 0.1 | Euclidean | $0.384 \pm 0.002$ | $0.424 \pm 0.002$ | $0.018 \pm 0.0$ | $0.026 \pm 0.0$ |
| Swiss roll high | 0.5 | Diffusion Map | $0.98 \pm 0.002$ | $0.985 \pm 0.002$ | $0.018 \pm 0.0$ | $0.026 \pm 0.0$ |
| Swiss roll high | 0.5 | Heat-Geo | $0.998 \pm 0.0$ | $0.997 \pm 0.0$ | $0.018 \pm 0.0$ | $0.026 \pm 0.0$ |
| Swiss roll high | 0.5 | Heat-PHATE | $0.995 \pm 0.0$ | $0.997 \pm 0.0$ | $0.018 \pm 0.0$ | $0.026 \pm 0.0$ |
| Swiss roll high | 0.5 | PHATE | $0.754 \pm 0.01$ | $0.756 \pm 0.006$ | $0.018 \pm 0.0$ | $0.026 \pm 0.0$ |
| Swiss roll high | 0.5 | Rand-Geo | $0.945 \pm 0.001$ | $0.945 \pm 0.002$ | $0.018 \pm 0.0$ | $0.026 \pm 0.0$ |
| Swiss roll high | 0.5 | Shortest Path | $\mathbf{0.999 \pm 0.0}$ | $\mathbf{0.998 \pm 0.0}$ | $0.018 \pm 0.0$ | $0.026 \pm 0.0$ |
| Swiss roll high | 0.5 | TSNE | $0.905 \pm 0.006$ | $0.899 \pm 0.003$ | $\mathbf{0.006 \pm 0.0}$ | $\mathbf{0.008 \pm 0.0}$ |
| Swiss roll high | 0.5 | UMAP | $0.876 \pm 0.017$ | $0.86 \pm 0.024$ | $0.012 \pm 0.0$ | $0.017 \pm 0.001$ |
| Swiss roll high | 0.5 | Euclidean | $0.38 \pm 0.003$ | $0.421 \pm 0.002$ | $0.018 \pm 0.0$ | $0.026 \pm 0.0$ |
| Swiss roll high | 1.0 | Diffusion Map | $0.555 \pm 0.155$ | $0.526 \pm 0.081$ | $0.018 \pm 0.0$ | $0.026 \pm 0.0$ |
| Swiss roll high | 1.0 | Heat-Geo | $0.643 \pm 0.173$ | $0.693 \pm 0.114$ | $0.018 \pm 0.0$ | $0.026 \pm 0.0$ |
| Swiss roll high | 1.0 | Heat-PHATE | $0.609 \pm 0.17$ | $0.611 \pm 0.121$ | $0.018 \pm 0.0$ | $0.026 \pm 0.0$ |
| Swiss roll high | 1.0 | PHATE | $0.271 \pm 0.025$ | $0.343 \pm 0.011$ | $0.018 \pm 0.0$ | $0.026 \pm 0.0$ |
| Swiss roll high | 1.0 | Rand-Geo | $0.41 \pm 0.038$ | $0.446 \pm 0.03$ | $0.018 \pm 0.0$ | $0.026 \pm 0.0$ |
| Swiss roll high | 1.0 | Shortest Path | $0.343 \pm 0.013$ | $0.4 \pm 0.007$ | $0.018 \pm 0.0$ | $0.026 \pm 0.0$ |
| Swiss roll high | 1.0 | TSNE | $0.737 \pm 0.124$ | $0.723 \pm 0.099$ | $\mathbf{0.008 \pm 0.001}$ | $\mathbf{0.015 \pm 0.003}$ |
| Swiss roll high | 1.0 | UMAP | $\mathbf{0.893 \pm 0.055}$ | $\mathbf{0.889 \pm 0.083}$ | $0.014 \pm 0.001$ | $0.02 \pm 0.001$ |
| Swiss roll high | 1.0 | Euclidean | $0.37 \pm 0.003$ | $0.414 \pm 0.002$ | $0.018 \pm 0.0$ | $0.026 \pm 0.0$ |
| Swiss roll high | 5.0 | Diffusion Map | $0.164 \pm 0.016$ | $0.174 \pm 0.009$ | $0.018 \pm 0.0$ | $0.026 \pm 0.0$ |
| Swiss roll high | 5.0 | Heat-PHATE | $\mathbf{0.202 \pm 0.01}$ | $\mathbf{0.236 \pm 0.009}$ | $0.018 \pm 0.0$ | $0.026 \pm 0.0$ |
| Swiss roll high | 5.0 | PHATE | $0.201 \pm 0.01$ | $0.234 \pm 0.008$ | $0.018 \pm 0.0$ | $0.026 \pm 0.0$ |
| Swiss roll high | 5.0 | Rand-Geo | $0.192 \pm 0.009$ | $0.228 \pm 0.008$ | $0.018 \pm 0.0$ | $0.026 \pm 0.0$ |
| Swiss roll high | 5.0 | Shortest Path | $0.187 \pm 0.01$ | $0.221 \pm 0.009$ | $0.018 \pm 0.0$ | $0.026 \pm 0.0$ |
| Swiss roll high | 5.0 | TSNE | $0.182 \pm 0.011$ | $0.213 \pm 0.01$ | $\mathbf{0.013 \pm 0.0}$ | $\mathbf{0.019 \pm 0.001}$ |
| Swiss roll high | 5.0 | UMAP | $0.195 \pm 0.009$ | $0.227 \pm 0.008$ | $0.016 \pm 0.0$ | $0.024 \pm 0.001$ |
| Swiss roll very high | 0.1 | Diffusion Map | $0.977 \pm 0.005$ | $0.984 \pm 0.004$ | $0.018 \pm 0.0$ | $0.026 \pm 0.0$ |
| Swiss roll very high | 0.1 | Heat-Geo | $0.996 \pm 0.001$ | $0.997 \pm 0.001$ | $0.018 \pm 0.0$ | $0.026 \pm 0.0$ |
| Swiss roll very high | 0.1 | Heat-PHATE | $0.991 \pm 0.001$ | $0.997 \pm 0.001$ | $0.018 \pm 0.0$ | $0.026 \pm 0.0$ |
| Swiss roll very high | 0.1 | PHATE | $0.683 \pm 0.023$ | $0.701 \pm 0.016$ | $0.018 \pm 0.0$ | $0.026 \pm 0.0$ |
| Swiss roll very high | 0.1 | Rand-Geo | $0.918 \pm 0.002$ | $0.917 \pm 0.002$ | $0.018 \pm 0.0$ | $0.026 \pm 0.0$ |
| Swiss roll very high | 0.1 | Shortest Path | $\mathbf{0.999 \pm 0.0}$ | $\mathbf{0.999 \pm 0.0}$ | $0.018 \pm 0.0$ | $0.026 \pm 0.0$ |
| Swiss roll very high | 0.1 | TSNE | $0.905 \pm 0.006$ | $0.897 \pm 0.004$ | $\mathbf{0.006 \pm 0.0}$ | $\mathbf{0.008 \pm 0.0}$ |
| Swiss roll very high | 0.1 | UMAP | $0.785 \pm 0.024$ | $0.781 \pm 0.017$ | $0.011 \pm 0.0$ | $0.016 \pm 0.001$ |
| Swiss roll very high | 0.1 | Euclidean | $0.384 \pm 0.002$ | $0.424 \pm 0.002$ | $0.018 \pm 0.0$ | $0.026 \pm 0.0$ |
| Swiss roll very high | 0.5 | Diffusion Map | $0.978 \pm 0.002$ | $0.984 \pm 0.001$ | $0.018 \pm 0.0$ | $0.026 \pm 0.0$ |
| Swiss roll very high | 0.5 | Heat-Geo | $0.997 \pm 0.0$ | $\mathbf{0.998 \pm 0.0}$ | $0.018 \pm 0.0$ | $0.026 \pm 0.0$ |
| Swiss roll very high | 0.5 | Heat-PHATE | $0.996 \pm 0.001$ | $0.997 \pm 0.0$ | $0.018 \pm 0.0$ | $0.026 \pm 0.0$ |
| Swiss roll very high | 0.5 | PHATE | $0.827 \pm 0.003$ | $0.815 \pm 0.002$ | $0.018 \pm 0.0$ | $0.026 \pm 0.0$ |
| Swiss roll very high | 0.5 | Rand-Geo | $0.944 \pm 0.001$ | $0.944 \pm 0.001$ | $0.018 \pm 0.0$ | $0.026 \pm 0.0$ |
| Swiss roll very high | 0.5 | Shortest Path | $\mathbf{0.998 \pm 0.0}$ | $0.997 \pm 0.0$ | $0.018 \pm 0.0$ | $0.026 \pm 0.0$ |
| Swiss roll very high | 0.5 | TSNE | $0.917 \pm 0.009$ | $0.917 \pm 0.007$ | $\mathbf{0.006 \pm 0.0}$ | $\mathbf{0.008 \pm 0.001}$ |
| Swiss roll very high | 0.5 | UMAP | $0.928 \pm 0.01$ | $0.929 \pm 0.012$ | $0.012 \pm 0.0$ | $0.017 \pm 0.001$ |
| Swiss roll very high | 0.5 | Euclidean | $0.379 \pm 0.002$ | $0.42 \pm 0.002$ | $0.018 \pm 0.0$ | $0.026 \pm 0.0$ |
| Swiss roll very high | 1.0 | Diffusion Map | $0.324 \pm 0.061$ | $0.399 \pm 0.033$ | $0.018 \pm 0.0$ | $0.026 \pm 0.0$ |
| Swiss roll very high | 1.0 | Heat-Geo | $0.364 \pm 0.008$ | $0.425 \pm 0.015$ | $0.018 \pm 0.0$ | $0.026 \pm 0.0$ |
| Swiss roll very high | 1.0 | Heat-PHATE | $0.352 \pm 0.022$ | $0.411 \pm 0.018$ | $0.018 \pm 0.0$ | $0.026 \pm 0.0$ |
| Swiss roll very high | 1.0 | PHATE | $0.326 \pm 0.009$ | $0.388 \pm 0.007$ | $0.018 \pm 0.0$ | $0.026 \pm 0.0$ |
| Swiss roll very high | 1.0 | Rand-Geo | $0.357 \pm 0.007$ | $0.404 \pm 0.005$ | $0.018 \pm 0.0$ | $0.026 \pm 0.0$ |
| Swiss roll very high | 1.0 | Shortest Path | $0.335 \pm 0.014$ | $0.39 \pm 0.011$ | $0.018 \pm 0.0$ | $0.026 \pm 0.0$ |
| Swiss roll very high | 1.0 | TSNE | $0.515 \pm 0.014$ | $0.522 \pm 0.01$ | $\mathbf{0.012 \pm 0.0}$ | $\mathbf{0.016 \pm 0.0}$ |
| Swiss roll very high | 1.0 | UMAP | $\mathbf{0.765 \pm 0.059}$ | $\mathbf{0.737 \pm 0.058}$ | $0.015 \pm 0.0$ | $0.021 \pm 0.0$ |
| Swiss roll very high | 1.0 | Euclidean | $0.366 \pm 0.002$ | $0.41 \pm 0.001$ | $0.018 \pm 0.0$ | $0.026 \pm 0.0$ |

Table 8: Comparison of the estimated distance matrices with the ground truth geodesic distance matrices on the Swiss roll dataset, using a two-dimensional embedding. Best models on average are bolded (not necessarily significant).

| data | Noise level | Method | PearsonR | SpearmanR | Norm Fro N2 | Norm inf N2 |
|---|---|---|---|---|---|---|
| Tree | 0.1 | Diffusion Map | $0.748 \pm 0.125$ | $0.733 \pm 0.111$ | $0.113 \pm 0.012$ | $0.161 \pm 0.019$ |
| Tree | 0.1 | Heat-Geo | $\mathbf{0.943 \pm 0.037}$ | $\mathbf{0.94 \pm 0.037}$ | $0.113 \pm 0.012$ | $0.161 \pm 0.019$ |
| Tree | 0.1 | Heat-PHATE | $0.872 \pm 0.04$ | $0.83 \pm 0.061$ | $0.113 \pm 0.012$ | $0.161 \pm 0.019$ |
| Tree | 0.1 | PHATE | $0.564 \pm 0.039$ | $0.469 \pm 0.052$ | $0.113 \pm 0.011$ | $0.161 \pm 0.018$ |
| Tree | 0.1 | Rand-Geo | $0.868 \pm 0.017$ | $0.85 \pm 0.019$ | $0.113 \pm 0.012$ | $0.161 \pm 0.019$ |
| Tree | 0.1 | Shortest Path | $0.937 \pm 0.037$ | $0.931 \pm 0.041$ | $0.113 \pm 0.012$ | $0.161 \pm 0.019$ |
| Tree | 0.1 | TSNE | $0.847 \pm 0.034$ | $0.824 \pm 0.045$ | $\mathbf{0.082 \pm 0.012}$ | $\mathbf{0.123 \pm 0.022}$ |
| Tree | 0.1 | UMAP | $0.692 \pm 0.058$ | $0.671 \pm 0.047$ | $0.107 \pm 0.012$ | $0.153 \pm 0.019$ |
| Tree | 0.1 | Euclidean | $0.809 \pm 0.017$ | $0.778 \pm 0.024$ | $0.113 \pm 0.012$ | $0.161 \pm 0.019$ |
| Tree | 0.5 | Diffusion Map | $0.656 \pm 0.054$ | $0.653 \pm 0.057$ | $0.113 \pm 0.012$ | $0.161 \pm 0.019$ |
| Tree | 0.5 | Heat-Geo | $\mathbf{0.806 \pm 0.019}$ | $\mathbf{0.787 \pm 0.009}$ | $0.113 \pm 0.012$ | $0.161 \pm 0.019$ |
| Tree | 0.5 | Heat-PHATE | $0.746 \pm 0.024$ | $0.744 \pm 0.031$ | $0.113 \pm 0.012$ | $0.161 \pm 0.019$ |
| Tree | 0.5 | PHATE | $0.766 \pm 0.023$ | $0.746 \pm 0.03$ | $0.113 \pm 0.011$ | $0.161 \pm 0.018$ |
| Tree | 0.5 | Rand-Geo | $0.721 \pm 0.024$ | $0.694 \pm 0.024$ | $0.113 \pm 0.012$ | $0.161 \pm 0.019$ |
| Tree | 0.5 | Shortest Path | $0.765 \pm 0.01$ | $0.738 \pm 0.011$ | $0.113 \pm 0.012$ | $0.161 \pm 0.019$ |
| Tree | 0.5 | TSNE | $0.795 \pm 0.046$ | $0.766 \pm 0.055$ | $\mathbf{0.083 \pm 0.012}$ | $\mathbf{0.128 \pm 0.018}$ |
| Tree | 0.5 | UMAP | $0.783 \pm 0.06$ | $0.757 \pm 0.054$ | $0.11 \pm 0.011$ | $0.157 \pm 0.018$ |
| Tree | 0.5 | Euclidean | $0.704 \pm 0.02$ | $0.672 \pm 0.038$ | $0.113 \pm 0.012$ | $0.161 \pm 0.019$ |
| Tree | 1.0 | Diffusion Map | $0.538 \pm 0.05$ | $0.471 \pm 0.089$ | $0.113 \pm 0.012$ | $0.161 \pm 0.019$ |
| Tree | 1.0 | Heat-Geo | $0.613 \pm 0.025$ | $\mathbf{0.58 \pm 0.036}$ | $0.113 \pm 0.012$ | $0.161 \pm 0.019$ |
| Tree | 1.0 | Heat-PHATE | $0.614 \pm 0.02$ | $0.571 \pm 0.044$ | $0.113 \pm 0.012$ | $0.161 \pm 0.019$ |
| Tree | 1.0 | PHATE | $\mathbf{0.615 \pm 0.017}$ | $0.572 \pm 0.036$ | $0.113 \pm 0.011$ | $0.161 \pm 0.018$ |
| Tree | 1.0 | Rand-Geo | $0.487 \pm 0.064$ | $0.465 \pm 0.071$ | $0.113 \pm 0.012$ | $0.161 \pm 0.019$ |
| Tree | 1.0 | Shortest Path | $0.542 \pm 0.047$ | $0.514 \pm 0.06$ | $0.113 \pm 0.012$ | $0.161 \pm 0.019$ |
| Tree | 1.0 | TSNE | $0.583 \pm 0.042$ | $0.553 \pm 0.045$ | $\mathbf{0.086 \pm 0.011}$ | $\mathbf{0.135 \pm 0.017}$ |
| Tree | 1.0 | UMAP | $0.595 \pm 0.032$ | $0.562 \pm 0.036$ | $0.111 \pm 0.011$ | $0.158 \pm 0.019$ |
| Tree | 1.0 | Euclidean | $0.502 \pm 0.051$ | $0.479 \pm 0.064$ | $0.113 \pm 0.012$ | $0.161 \pm 0.019$ |
| Tree high | 0.1 | Diffusion Map | $0.754 \pm 0.049$ | $0.741 \pm 0.057$ | $0.267 \pm 0.021$ | $0.369 \pm 0.026$ |
| Tree high | 0.1 | Heat-Geo | $0.956 \pm 0.014$ | $\mathbf{0.957 \pm 0.015}$ | $0.267 \pm 0.021$ | $0.369 \pm 0.026$ |
| Tree high | 0.1 | Heat-PHATE | $0.831 \pm 0.082$ | $0.764 \pm 0.115$ | $0.267 \pm 0.021$ | $0.369 \pm 0.026$ |
| Tree high | 0.1 | PHATE | $0.484 \pm 0.036$ | $0.4 \pm 0.028$ | $0.267 \pm 0.02$ | $0.369 \pm 0.025$ |
| Tree high | 0.1 | Rand-Geo | $0.817 \pm 0.013$ | $0.774 \pm 0.022$ | $0.267 \pm 0.021$ | $0.369 \pm 0.026$ |
| Tree high | 0.1 | Shortest Path | $\mathbf{0.958 \pm 0.014}$ | $0.956 \pm 0.017$ | $0.267 \pm 0.021$ | $0.369 \pm 0.026$ |
| Tree high | 0.1 | TSNE | $0.89 \pm 0.039$ | $0.866 \pm 0.043$ | $\mathbf{0.233 \pm 0.021}$ | $\mathbf{0.327 \pm 0.026}$ |
| Tree high | 0.1 | UMAP | $0.8 \pm 0.031$ | $0.764 \pm 0.034$ | $0.259 \pm 0.021$ | $0.36 \pm 0.028$ |
| Tree high | 0.1 | Euclidean | $0.878 \pm 0.042$ | $0.859 \pm 0.051$ | $0.267 \pm 0.021$ | $0.369 \pm 0.026$ |
| Tree high | 0.5 | Diffusion Map | $0.706 \pm 0.124$ | $0.705 \pm 0.113$ | $0.267 \pm 0.021$ | $0.369 \pm 0.026$ |
| Tree high | 0.5 | Heat-Geo | $\mathbf{0.932 \pm 0.022}$ | $\mathbf{0.928 \pm 0.023}$ | $0.267 \pm 0.021$ | $0.369 \pm 0.026$ |
| Tree high | 0.5 | Heat-PHATE | $0.923 \pm 0.023$ | $0.921 \pm 0.022$ | $0.267 \pm 0.021$ | $0.369 \pm 0.026$ |
| Tree high | 0.5 | PHATE | $0.844 \pm 0.048$ | $0.79 \pm 0.07$ | $0.267 \pm 0.02$ | $0.369 \pm 0.025$ |
| Tree high | 0.5 | Rand-Geo | $0.875 \pm 0.042$ | $0.855 \pm 0.048$ | $0.267 \pm 0.021$ | $0.369 \pm 0.026$ |
| Tree high | 0.5 | Shortest Path | $0.917 \pm 0.025$ | $0.91 \pm 0.03$ | $0.267 \pm 0.021$ | $0.369 \pm 0.026$ |
| Tree high | 0.5 | TSNE | $0.922 \pm 0.035$ | $0.91 \pm 0.045$ | $\mathbf{0.237 \pm 0.021}$ | $\mathbf{0.334 \pm 0.027}$ |
| Tree high | 0.5 | UMAP | $0.823 \pm 0.054$ | $0.803 \pm 0.041$ | $0.261 \pm 0.021$ | $0.361 \pm 0.026$ |
| Tree high | 0.5 | Euclidean | $0.819 \pm 0.048$ | $0.799 \pm 0.053$ | $0.267 \pm 0.021$ | $0.369 \pm 0.026$ |
| Tree high | 1.0 | Diffusion Map | $0.598 \pm 0.117$ | $0.613 \pm 0.103$ | $0.267 \pm 0.021$ | $0.369 \pm 0.026$ |
| Tree high | 1.0 | Heat-Geo | $0.794 \pm 0.066$ | $0.805 \pm 0.049$ | $0.267 \pm 0.021$ | $0.369 \pm 0.026$ |
| Tree high | 1.0 | Heat-PHATE | $0.826 \pm 0.064$ | $0.823 \pm 0.067$ | $0.267 \pm 0.021$ | $0.369 \pm 0.026$ |
| Tree high | 1.0 | PHATE | $0.827 \pm 0.059$ | $0.82 \pm 0.062$ | $0.267 \pm 0.02$ | $0.369 \pm 0.025$ |
| Tree high | 1.0 | Rand-Geo | $0.71 \pm 0.043$ | $0.686 \pm 0.045$ | $0.267 \pm 0.021$ | $0.369 \pm 0.026$ |
| Tree high | 1.0 | Shortest Path | $0.771 \pm 0.064$ | $0.753 \pm 0.07$ | $0.267 \pm 0.021$ | $0.369 \pm 0.026$ |
| Tree high | 1.0 | TSNE | $0.84 \pm 0.066$ | $0.821 \pm 0.074$ | $\mathbf{0.238 \pm 0.02}$ | $\mathbf{0.335 \pm 0.026}$ |
| Tree high | 1.0 | UMAP | $\mathbf{0.853 \pm 0.051}$ | $\mathbf{0.839 \pm 0.057}$ | $0.264 \pm 0.021$ | $0.365 \pm 0.026$ |
| Tree high | 1.0 | Euclidean | $0.683 \pm 0.067$ | $0.665 \pm 0.07$ | $0.267 \pm 0.021$ | $0.369 \pm 0.026$ |

Table 9: Comparison of the estimated distance matrices with the ground truth geodesic distance matrices on the Tree dataset, using a two-dimensional embedding. Best models on average are bolded (not necessarily significant).

| data | Noise level | Method | Homogeneity | Adjusted Rand Score | Adjusted Mutual Info Score |
|---|---|---|---|---|---|
| Swiss roll | 0.1 | Heat-Geo | **0.82 ± 0.008** | **0.668 ± 0.034** | **0.74 ± 0.018** |
| Swiss roll | 0.1 | Phate | 0.731 ± 0.035 | 0.546 ± 0.044 | 0.652 ± 0.046 |
| Swiss roll | 0.1 | TSNE | 0.748 ± 0.067 | 0.537 ± 0.1 | 0.668 ± 0.068 |
| Swiss roll | 0.1 | UMAP | 0.81 ± 0.036 | 0.611 ± 0.039 | 0.726 ± 0.045 |
| Swiss roll | 0.5 | Heat-Geo | 0.813 ± 0.026 | 0.656 ± 0.049 | 0.733 ± 0.022 |
| Swiss roll | 0.5 | Phate | 0.735 ± 0.048 | 0.543 ± 0.064 | 0.656 ± 0.053 |
| Swiss roll | 0.5 | TSNE | 0.764 ± 0.07 | 0.564 ± 0.097 | 0.684 ± 0.065 |
| Swiss roll | 0.5 | UMAP | **0.826 ± 0.019** | **0.664 ± 0.073** | **0.744 ± 0.032** |
| Swiss roll | 1.0 | Heat-Geo | 0.722 ± 0.051 | 0.548 ± 0.091 | 0.652 ± 0.056 |
| Swiss roll | 1.0 | Phate | 0.482 ± 0.014 | 0.317 ± 0.031 | 0.428 ± 0.021 |
| Swiss roll | 1.0 | TSNE | **0.757 ± 0.037** | **0.562 ± 0.058** | **0.679 ± 0.042** |
| Swiss roll | 1.0 | UMAP | 0.726 ± 0.041 | 0.51 ± 0.077 | 0.65 ± 0.05 |
| Swiss roll high | 0.1 | Heat-Geo | **0.82 ± 0.015** | **0.666 ± 0.033** | **0.739 ± 0.019** |
| Swiss roll high | 0.1 | Phate | 0.705 ± 0.03 | 0.518 ± 0.048 | 0.628 ± 0.04 |
| Swiss roll high | 0.1 | TSNE | 0.757 ± 0.078 | 0.558 ± 0.115 | 0.677 ± 0.08 |
| Swiss roll high | 0.1 | UMAP | 0.796 ± 0.03 | 0.624 ± 0.048 | 0.714 ± 0.037 |
| Swiss roll high | 0.5 | Heat-Geo | **0.805 ± 0.021** | **0.655 ± 0.047** | **0.725 ± 0.035** |
| Swiss roll high | 0.5 | Phate | 0.745 ± 0.04 | 0.562 ± 0.061 | 0.664 ± 0.047 |
| Swiss roll high | 0.5 | TSNE | 0.747 ± 0.075 | 0.538 ± 0.11 | 0.668 ± 0.075 |
| Swiss roll high | 0.5 | UMAP | 0.787 ± 0.041 | 0.573 ± 0.067 | 0.703 ± 0.032 |
| Swiss roll high | 1.0 | Heat-Geo | 0.7 ± 0.045 | 0.534 ± 0.057 | 0.644 ± 0.032 |
| Swiss roll high | 1.0 | Phate | 0.552 ± 0.047 | 0.386 ± 0.056 | 0.496 ± 0.04 |
| Swiss roll high | 1.0 | TSNE | 0.754 ± 0.034 | 0.548 ± 0.068 | 0.675 ± 0.036 |
| Swiss roll high | 1.0 | UMAP | **0.76 ± 0.041** | **0.56 ± 0.077** | **0.68 ± 0.05** |
| Swiss roll very high | 0.1 | Heat-Geo | **0.818 ± 0.033** | **0.668 ± 0.074** | **0.738 ± 0.039** |
| Swiss roll very high | 0.1 | Phate | 0.688 ± 0.043 | 0.497 ± 0.053 | 0.614 ± 0.053 |
| Swiss roll very high | 0.1 | TSNE | 0.741 ± 0.07 | 0.544 ± 0.101 | 0.662 ± 0.075 |
| Swiss roll very high | 0.1 | UMAP | 0.816 ± 0.042 | 0.65 ± 0.069 | 0.733 ± 0.054 |
| Swiss roll very high | 0.5 | Heat-Geo | 0.73 ± 0.045 | **0.605 ± 0.093** | 0.701 ± 0.028 |
| Swiss roll very high | 0.5 | Phate | 0.758 ± 0.034 | 0.55 ± 0.037 | 0.676 ± 0.014 |
| Swiss roll very high | 0.5 | TSNE | 0.77 ± 0.054 | 0.557 ± 0.093 | **0.708 ± 0.031** |
| Swiss roll very high | 0.5 | UMAP | **0.789 ± 0.052** | 0.574 ± 0.101 | 0.707 ± 0.061 |
| Swiss roll very high | 1.0 | Heat-Geo | 0.592 ± 0.033 | 0.427 ± 0.063 | 0.545 ± 0.031 |
| Swiss roll very high | 1.0 | Phate | 0.531 ± 0.042 | 0.377 ± 0.046 | 0.486 ± 0.045 |
| Swiss roll very high | 1.0 | TSNE | **0.738 ± 0.019** | **0.551 ± 0.039** | **0.662 ± 0.025** |
| Swiss roll very high | 1.0 | UMAP | 0.736 ± 0.057 | 0.542 ± 0.102 | 0.66 ± 0.061 |
| Tree | 0.1 | Heat-Geo | **0.784 ± 0.051** | **0.734 ± 0.07** | **0.786 ± 0.051** |
| Tree | 0.1 | Phate | 0.55 ± 0.042 | 0.409 ± 0.064 | 0.555 ± 0.042 |
| Tree | 0.1 | TSNE | 0.706 ± 0.054 | 0.61 ± 0.075 | 0.712 ± 0.055 |
| Tree | 0.1 | UMAP | 0.678 ± 0.086 | 0.584 ± 0.12 | 0.681 ± 0.086 |
| Tree | 0.5 | Heat-Geo | 0.545 ± 0.121 | 0.411 ± 0.154 | 0.577 ± 0.094 |
| Tree | 0.5 | Phate | 0.529 ± 0.111 | 0.404 ± 0.151 | 0.555 ± 0.095 |
| Tree | 0.5 | TSNE | **0.647 ± 0.049** | **0.591 ± 0.065** | 0.65 ± 0.048 |
| Tree | 0.5 | UMAP | 0.645 ± 0.051 | 0.565 ± 0.058 | **0.652 ± 0.05** |
| Tree | 1.0 | Heat-Geo | 0.398 ± 0.07 | 0.3 ± 0.077 | 0.42 ± 0.07 |
| Tree | 1.0 | Phate | 0.418 ± 0.08 | 0.337 ± 0.093 | 0.43 ± 0.075 |
| Tree | 1.0 | TSNE | 0.405 ± 0.077 | 0.378 ± 0.074 | 0.405 ± 0.077 |
| Tree | 1.0 | UMAP | **0.432 ± 0.086** | **0.395 ± 0.098** | **0.432 ± 0.085** |

Table 10: Clustering results on swiss roll (with distribution) and tree. Best models on average are bolded (not necessarily significant).

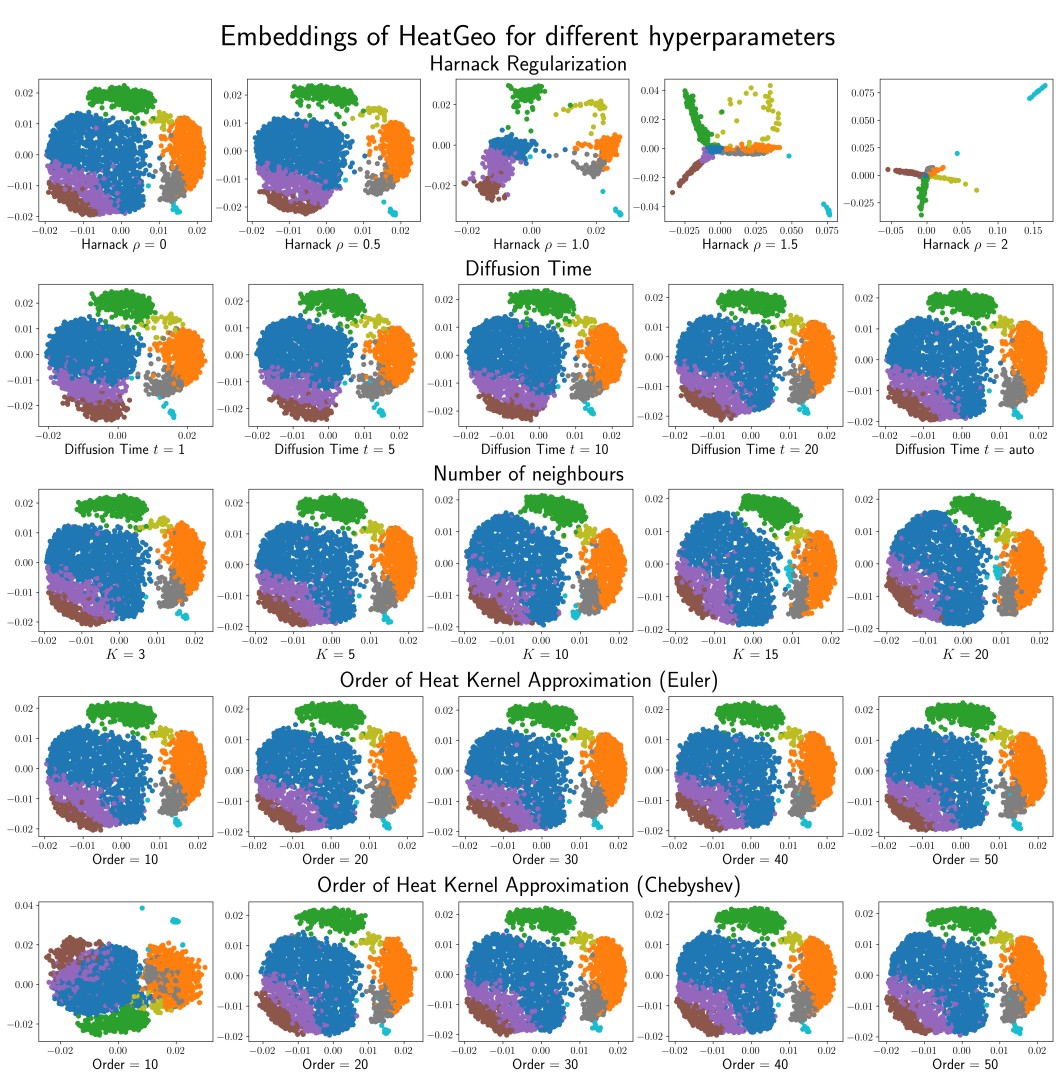

Figure 8: Embeddings of Heat Geodesic Embedding for different choices of hyperparameters on the EB dataset. We evaluate the impact of the Harnack regularization, the diffusion time, the number of neighbours in the kNN, and the order of the approximation for Euler and Checbyshev approximations.

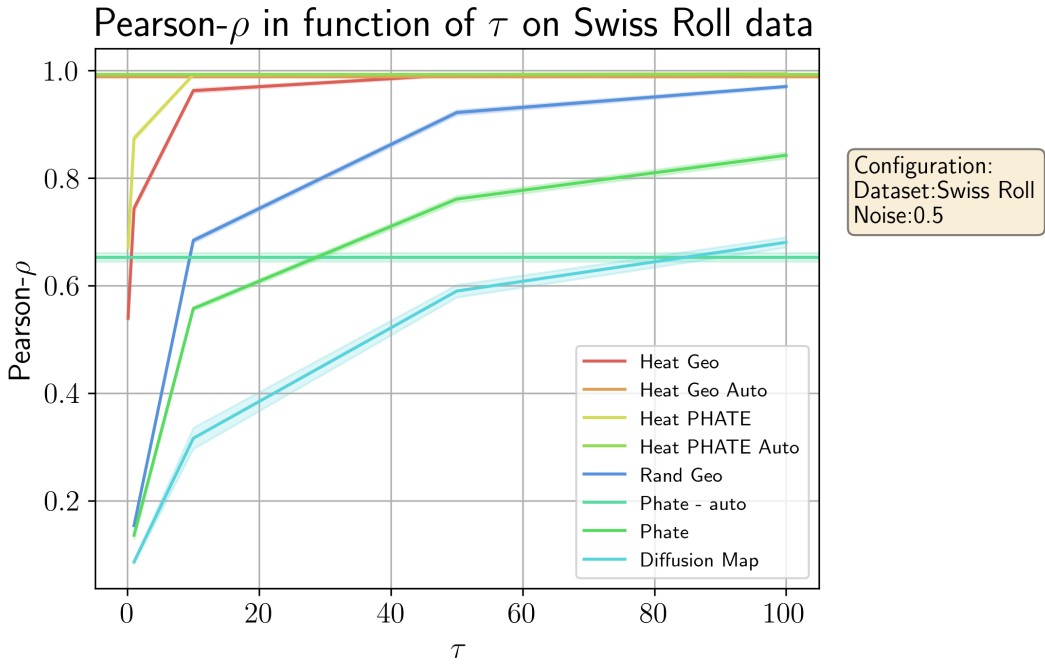

Figure 9: Impact of diffusion time on the Pearson correlation between the estimated distance matrix and ground truth distance matrix for different methods on the Swiss roll dataset.

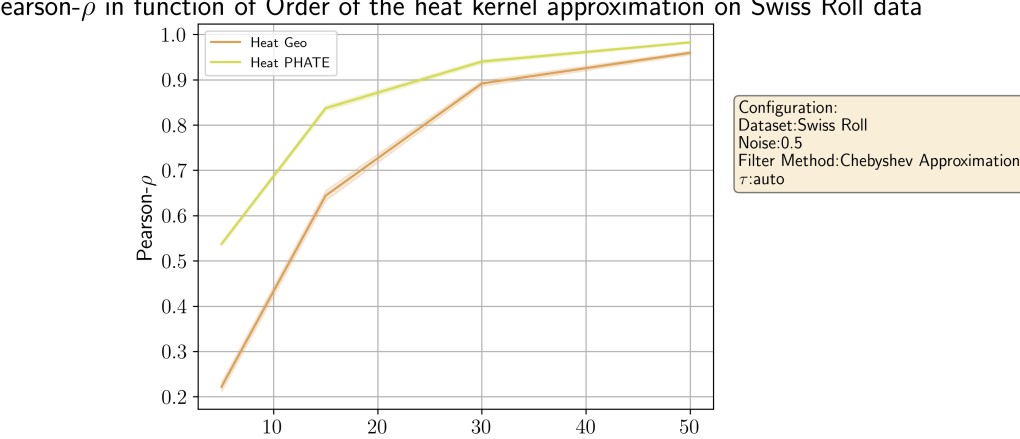

Figure 10: Impact of Checbyshev approximation order on the Pearson correlation between the estimated distance matrix and ground truth distance matrix for different methods on the Swiss roll dataset.

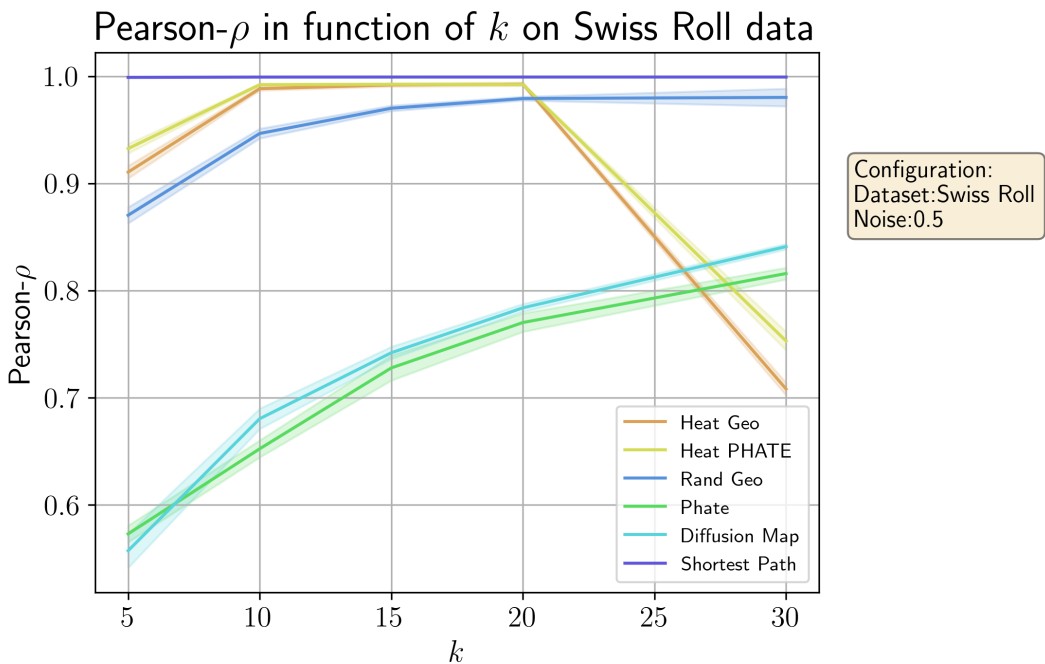

Figure 11: Impact of number of neighbours on the Pearson correlation between the estimated distance matrix and ground truth distance matrix for different methods on the Swiss roll dataset.

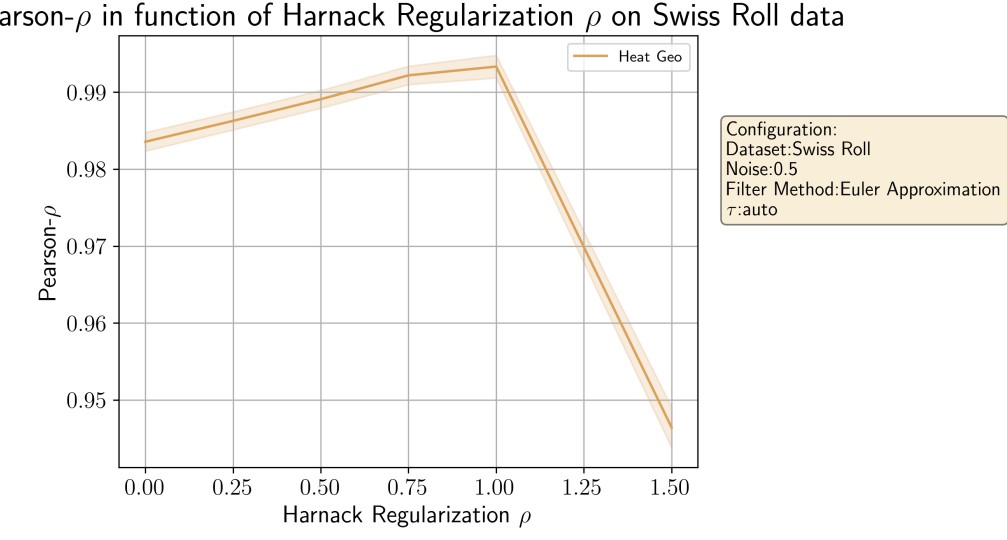

Figure 12: Impact of Harnack regularization on the Pearson correlation between the estimated distance matrix and ground truth distance matrix for HeatGeo on the Swiss roll dataset.

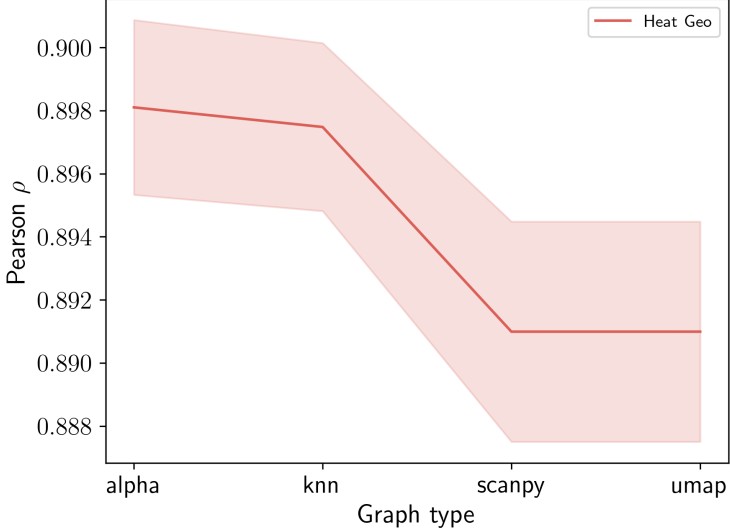

Figure 13: Pearson correlation between estimated and ground truth distance matrices for the 10-dimensional Swiss roll dataset for various graph constructions. Standard deviations are computed over the 5 test folds.

Table 11: Clustering quality metrics for different methods. We report the homogeneity and the adjusted mutual information (aMI). Best models on average are bolded (higher is better).

| data | Noise level | Method | Homogeneity | Adjusted Rand Score | Adjusted Mutual Info Score |
|------|-------------|--------|-------------|---------------------|----------------------------|
| Mnist | 0 | Diff-Map | $0.556 \pm 0.002$ | $0.347 \pm 0.002$ | $0.622 \pm 0.002$ |
| Mnist | 0 | Heat-Geo | $0.785 \pm 0.0$ | $0.695 \pm 0.0$ | $0.829 \pm 0.001$ |
| Mnist | 0 | Phate | $0.822 \pm 0.01$ | $0.72 \pm 0.017$ | $0.835 \pm 0.011$ |
| Mnist | 0 | TSNE | $\mathbf{0.903 \pm 0.003}$ | $\mathbf{0.871 \pm 0.002}$ | $\mathbf{0.902 \pm 0.003}$ |
| Mnist | 0 | UMAP | $0.851 \pm 0.016$ | $0.846 \pm 0.005$ | $0.86 \pm 0.015$ |
| Coil | 0 | Diff-Map | $0.21 \pm 0.036$ | $0.041 \pm 0.015$ | $0.142 \pm 0.024$ |
| Coil | 0 | Heat-Geo | $0.849 \pm 0.016$ | $0.67 \pm 0.029$ | $0.806 \pm 0.022$ |
| Coil | 0 | Phate | $0.804 \pm 0.017$ | $0.615 \pm 0.028$ | $0.735 \pm 0.021$ |
| Coil | 0 | TSNE | $\mathbf{0.907 \pm 0.014}$ | $\mathbf{0.79 \pm 0.03}$ | $\mathbf{0.88 \pm 0.02}$ |
| Coil | 0 | UMAP | $0.871 \pm 0.009$ | $0.725 \pm 0.019$ | $0.826 \pm 0.012$ |
| Pbmc | 0 | Diff-Map | $0.026 \pm 0.001$ | $0.011 \pm 0.0$ | $0.038 \pm 0.001$ |
| Pbmc | 0 | Heat-Geo | $0.734 \pm 0.009$ | $0.724 \pm 0.019$ | $0.768 \pm 0.017$ |
| Pbmc | 0 | Phate | $\mathbf{0.798 \pm 0.012}$ | $\mathbf{0.818 \pm 0.009}$ | $\mathbf{0.785 \pm 0.01}$ |
| Pbmc | 0 | TSNE | $0.605 \pm 0.019$ | $0.437 \pm 0.032$ | $0.544 \pm 0.022$ |
| Pbmc | 0 | UMAP | $0.177 \pm 0.037$ | $0.097 \pm 0.033$ | $0.148 \pm 0.035$ |

Table 12: Average computing time for creating embeddings and corresponding distance matrix for the different methods. All methods are applied on the Swiss roll dataset in three dimension with 2000 samples. We present empirical averages and standard deviations over ten repetitions.

| Method | Time (s) |
|--------|----------|
| UMAP | $4.21 \pm 0.68$ |
| t-SNE | $80.15 \pm 62.44$ |
| Isomap | $2.58 \pm 0.03$ |
| PHATE | $3.30 \pm 0.05$ |
| Diff Map | $8.62 \pm 0.23$ |
| HeatGeo (Backward Euler) | $2.60 \pm 0.05$ |
| HeatGeo (Chebyshev) | $2.11 \pm 0.05$ |

