}$, we have for triples $x, y, z \in \boldsymbol{X}$ that $|x - y| > |x - z|$ implies $d_t(x, y) > d_t(x, z)$, i.e. the heat-geodesic dissimilarity is order preserving.*

*Proof.* When $\sigma = 0$ or the manifold has uniform volume growth we need only consider the $-4t \log(\boldsymbol{H}_t)_{ij}$ terms. The assumption that $|x - y| > |x - z|$ implies $\boldsymbol{H}_t(x, y) < \boldsymbol{H}_t(x, z)$. We are able to conclude that $-4t \log \boldsymbol{H}_t(x, y) > -4t \log \boldsymbol{H}_t(x, z)$ and thus $d_t(x, y) > d_t(x, z)$. $\