# OpenReview forum: "A Heat Diffusion Perspective on Geodesic Preserving Dimensionality Reduction"
_NeurIPS.cc/2023/Conference — NeurIPS 2023 poster_

### Official Review · Reviewer_Yhnd · 2023-06-16

**Soundness:** 2 fair
**Presentation:** 2 fair
**Contribution:** 3 good
**Rating:** 5
**Confidence:** 3

**Summary:**

In the paper "A heat diffusion perspective on geodesic preserving dimensionality reduction", the authors develop a new embedding method (HeatGeo), conceptually close to PHATE (Moon et al 2019) in that it performs MDS on some kind of diffusion distances. The authors make some strong claims about HeatGeo: (1) it is SOTA in preserving manifold distances; (2) it is SOTA in preserving cluster structures; (3) it can approximate both PHATE and t-SNE depending on parameters.

I did not fully understand/follow all the math, but I found the paper nevertheless interesting as it gives a systematic/principled treatment of heat diffusion MDS methods. That said, I was not entirely convinced by the evaluations and especially claims (2) and (3) seem to me too strong and IMHO not sufficiently supported by the shown evidence. Also, many places were not really clear to me. Overall I am somewhat on the fence about this paper: it seems it is not fully ready yet. I am giving it a borderline score.

**Strengths:**

* Systematic/principled mathematical treatment (which admittedly I did not fully grasp)
* Benchmarks of several embedding methods on artificial datasets
* Applications to real RNAseq data
* Relationships between HeatGeo and existing methods (PHATE/t-SNE)

**Weaknesses:**

* Relationship to PHATE is not very clear
* Very bold claims that are not fully supported by the evidence
* Insufficient evaluation, in particular many embeddings are not shown
* Runtime is not discussed

Now in more detail:

* In spirit, the method seems very similar to PHATE. First, kNN graph is used to construct NxN distance matrix between all points (using heat diffusion). Second, MDS is used to produce the embedding. Of course there are many small differences from PHATE in how exactly all of this is setup up, and the way this paper is written, it comes across as more mathematically principled than PHATE. But in the end, I was still confused about what is the MAIN conceptual/algorithmic difference. That's very important in order to judge the relative novelty of HeatGeo. Is it conceptually a minor tweak to / simplification of PHATE, or is it conceptually a very different thing?

* One bold claim is that HeatGeo outperforms other methods in preserving clusters (supported by Table 2). This is conceptually surprising to me, because I expect some trade-off between preserving geodesic distances and preserving clusters (similar trade-off was discussed in Ref [2]: https://jmlr.org/papers/v23/21-0055.html), so it's odd if the same method (HeatGeo) wins in both competitions. Also, we know that methods like t-SNE and UMAP really excel in representing clusters in high-dim data (just try them on MNIST) but more geodesic/MDS methods like Isomap or PHATE perform worse (again, just try them on MNIST).

  To be more convinced here, I would like to see the embeddings corresponding to Table 2. Also, I don't think "Tree" is a good dataset for clustering because we don't expect k-means to pick up the tree branches: just look at HeatGeo if Tree in Figure 1 -- How can k-means separate the branches? I don't really understand why HeatGeo performs well here in Table 2. Rather than the Tree, I would like to have some toy dataset with 100D Gaussians differently far away from each other. I would also love to see MNIST (a subset of MNIST).

* Another bold claim is that HeatGeo approximates SNE/t-SNE. I was very surprised to read that, because I have never seen SNE/t-SNE formulated as an MDS problem, and assumed it is impossible. However, I don't really see much evidence that this is true. Can you do HeatGeo of MNIST that would look like t-SNE? In the EB dataset (Figure 3), can you change HeatGeo params to make it look like t-SNE? The same for the PBMC dataset (Figure 4)?

* Figure 1: This is the main motivational figure showing that HeatGeo is very good at unrolling Swiss roll, whereas t-SNE creates "artifical clusters". Personally, I think it does not make much sense to apply t-SNE to an intrinsically 2D dataset because t-SNE is designed for reducing dimensionality (that's why it uses t-distribution kernel, unlike SNE). When working with intrinsically 2D data, it makes much more sense to use SNE instead of t-SNE. And indeed, SNE is much better at unrolling Swiss roll than t-SNE (example: https://jlmelville.github.io/smallvis/swisssne.html). As the authors discuss SNE in their paper anyway, I think it would be good to add SNE to the comparison tables and also to Figure 1 (can be done e.g. via openTSNE). By the way, Isomap can also unroll the Swiss roll perfectly.

* As the HeatGeo relies on MDS, it constructs a dense NxN distance matrix and so has O(N^2) complexity. All the datasets used in the paper are very small. This is unlike t-SNE/UMAP which can be run on tens of millions of points easily. I think this has to be discussed, e.g. in Section 7.


SMALLER POINTS AND QUESTIONS

* line 124: classic MDS does not optimize objective (5).

* line 148: Ref [2] shows that UMAP's effective loss function is very different from its stated loss function, so this formula is actually misleading. See also https://proceedings.neurips.cc/paper/2021/hash/2de5d16682c3c35007e4e92982f1a2ba-Abstract.html and https://openreview.net/forum?id=B8a1FcY0vi which develop this argument.

* line 194: "small t can lead to disconnected graphs" -- I don't understand this sentence, I thought the graph does not depend on t, no? You first construct the graph and the laplacian L, and only later you use the diffusion time t.

* line 211: Algorithm 1 -- I don't really understand how L is constructed. What is the underlying graph? Is it a kNN graph, what are the weights? What is the k value (number of neighbors)? I think this is not really spelled out in the text. PHATE used some complicated setup with alpha-kernels, are you also using this here?

* Table 1: this table is missing t-SNE and UMAP rows. Also, I would like to see SNE in there, and Isomap.

* Figure 3 caption: "UMAP and t-SNE do not capture the continuous manifold" -- actually UMAP looks pretty similar to HeatGeo to me. Also, Ref [2] argues that higher attraction in t-SNE leads to better manifold preservation, so UMAP is better at showing the manifold (which is what we see in your Fig 3) but one can do even better by using higher exaggeration in t-SNE. Maybe worth mentioning.

* line 314: "HeatGeo interpolated to PHATE for \rho->1" -- I was confused by that. What is rho? It wasn't really discussed before, I don't see it in section 5 that discusses relation to PHATE. I do see \rho in Algorithm 1 but not in the text of Section 4. So it's mysterious.

* Figure 4 does not show t-SNE, but Figure 5 in the Appendix does. I am surprised by how bad t-SNE looks. I have seen t-SNE of various PBMC RNAseq datasets many times, and t-SNE is usually very good at separating cell types. But in your appendix figure t-SNE looks very poor. Why is that?

**Questions:**

See above

**Limitations:**

See above

---

> ### Author Rebuttal · Authors · 2023-08-10
>
> We would like to thank the reviewer for their thorough analysis of our manuscript. In the following we will address the main points raised by the reviewers. Typos or undefined notation were fixed.
>
> **Relationship to PHATE is not very clear.** Our method follows the same main steps as Isomap and PHATE; 1) Graph construction, 2) Distance computation 3) MDS. Hence algorithmically, these methods are similar. Amongst the differences with PHATE is the use of the continuous dynamic (the heat kernel), the volume regularization, decoupling of the denoising. As you mentioned a key contribution is connecting diffusion-based distances to heat flow on the manifold, providing explicit bounds on the manifold geodesic distance. Thus we showcase a template that uses the framework of ISOMAP and PHATE but utilizes heat diffusion directly.
>
> **HeatGeo and clustering.** We would like to point the reviewer to the global response where we address the HeatGeo and clustering. (1) We believe that distance preservation is not at odds with the cluster structure preservation; (2) we used a mixture of Gaussian on the swiss roll (a combination of manifold and cluster); (3) we added evaluation on COIL-20 and MNIST.
>
> >Figure 1: This is the main motivational figure showing that HeatGeo is very good at unrolling Swiss roll…
>
> **On Figure 1.**  Thank you for providing us with this link, indeed it is a great example, however the results using 3k observations are not really convincing. In Fig. 1, we wanted to present results from the methods that are the most used in practice (although diffusion maps is not the most popular, we added it since it is another diffusion-based method). Further, for Table 2, we used different hyper-parameters specific to each method and selected them according to their performance on a validation set. We also used multiple repetitions to assess the variability of performance, while Fig.1 only shows one experiment. That said, Fig. 1 also clearly shows that UMAP and t-SNE don’t perform well on the tree dataset. The orange class is split on UMAP and the blue class is heavily distorted on the t-SNE representation. To compare the methods, we think that the quantitative results are much more powerful, in particular Tab.1 on distance preservation of the manifold. We have updated our comparisons, and now provide quantitative results for Isomap and metric-MDS. More details about the experiment setup is available at the beginning of Section 6, and the code is available [here](https://anonymous.4open.science/r/anon-heatgeo-CE2A/cluster_pred.py).
>
> **HeatGeo approximates SNE/t-SNE.** We would like to emphasize that our method can be reconfigured to be similar to a modification of SNE. We refer to proposition 4.3 for the details on the configuration. It is not true that HeatGeo directly approximates SNE, we will review the manuscript to make sure this statement is clear. We think the similarity between the two methods is clearer from a theoretical perspective. If we have space, we will move parts of the proof of prop. 4.3 to the main body of the text. The main point of this proof is that because the heat-geodesic dissimilarly uses the logarithm we can frame it as minimizing a KL loss (similar to SNE). The proof is provided in the supplementary material in section A.2.
>
> **On the graph construction.** The Laplacian is constructed from an affinity kernel. It is clearer in the full Alg.2 provided in the supplement, we will clarify it in the main body of the text. In our experiments we used the alpha-decay kernel to construct the graph. We experimented with other kernels and did not see a significant difference in accuracy. For example, in Fig. 13 we report the accuracy on the swiss roll for 4 types of kernels.
>
> **Complexity of HeatGeo.** Please refer to the global response for our comment on the algorithm’s complexity.
>
> > line 124: classic MDS does not optimize objective (5).
>
> You are correct, we use the metric MDS problem defined in [1], we made it clearer in the manuscript.
>
> >Table 1: this table is missing t-SNE and UMAP rows. Also, I would like to see SNE in there, and Isomap.
>
> In Tab.1 we compare the distance matrix used by a method against the ground truth distance matrix. UMAP, SNE, and t-SNE, do not rely on a distance matrix, but rather on an affinity kernel. So we cannot compare with them. We did a comparison against the shortest path distance used in Isomap, we noted it “Shortest path”. In this table, the focus is on the distances, not the embedding.
>
> >line 148: Ref [2] shows that UMAP's effective loss function…
>
> We modified our statement and added the provided citations.
>
> >line 194: "small t can lead to disconnected graphs"...
>
> This is a typo. The number of neighbors could cause a disconnected graph. We fixed it.
>
> > Figure 3 caption: "UMAP and t-SNE do not capture the continuous manifold"...
>
> We agree that there is not a visually sharp distinction between the two. We modified the caption of Fig.3 accordingly. High exaggeration in t-SNE is indeed interesting! We have added a footnote to this effect.
>
> >I am surprised by how bad t-SNE looks...
>
> For the clustering experiment, we try perplexity parameters of ​​10,30,100, and the figure is with perplexity 40, we are happy to try other parameters. But we did not observe that tSNE is good at separating this PBMC dataset. We believe there exist parameters where t-SNE performs well on PBMC, but it may take more work to find those.
>
> We thank the reviewer for their valuable feedback and great questions. We hope that our rebuttal fully addresses all their important salient points, and kindly ask the reviewer to potentially upgrade their score if the reviewer is satisfied with our responses. We are also happy to answer any further questions that arise.
>
> [1] Zheng, J. X. et al. (2018). Graph drawing by stochastic gradient descent. TVCG
>
> [2] Moon, K. R. et al. (2019). Visualizing structure and transitions in high-dimensional biological data. Nature biotechnology.

---

> > ### Comment · Reviewer_Yhnd · 2023-08-14
> > **Thank you**
> >
> > I thank the authors for the detailed rebuttal to all the reviews. I am increasing my score from 4 to 5.

---

> > > ### Author Response · Authors · 2023-08-15
> > > **Thank you !**
> > >
> > > Dear Reviewer Yhnd,
> > >
> > > Thank you for your positive feedback! We did not mean to present our method outside the scope of its capabilities and we will make sure to revise the manuscript to clearly outline its limitations.
> > >
> > > In particular, we have clarified the description of the clusters experiments in Section 6.2 and added a discussion between clusters preservation and geodesic distance preservation. We further added a discussion on the superior performance of t-SNE and UMAP on COIL and MNIST.
> > >
> > > Please let us know if we can provide any additional information to improve your remaining concerns.
> > >
> > > Thanks a lot,
> > >
> > > The authors

---

### Official Review · Reviewer_rWBe · 2023-07-07

**Soundness:** 4 excellent
**Presentation:** 4 excellent
**Contribution:** 3 good
**Rating:** 8
**Confidence:** 4

**Summary:**

The authors propose heat geodesic embeddings, a novel approach to perform graph embeddings that is motivated by Riemannian geometry (Varadhan's formula and the parabolic Harnack inequality), which allows for directly linking heat diffusion on graphs to manifold distances. They performed experiments to demonstrate the efficacy of their approach.

**Strengths:**

- Originality: The proposed heat geodesic embedding and the approach is , to the best of my knowledge, original and novel.

- Significance: It is certainly important to study spectral embeddings nowadays, and I find the authors' methodological contributions to be significant.

- Quality: The quality of the theoretical derivations and experimental evaluations is sound, to the best of my knowledge.

- Clarity: Despite the advanced mathematical material involved, the authors presented and motivated their approach in a clear and convincing manner.

**Weaknesses:**

- there are no obvious weaknesses in the paper. The method is presented well and is straightforward. The authors performed experiments which covered most of the bases.

**Questions:**

- see sections above.

**Limitations:**

- no negative societal impact.

---

> ### Author Rebuttal · Authors · 2023-08-10
>
> We are pleased to read such an encouraging perspective on our work and thank the reviewer for their feedback. We remain at their disposal for any further clarification.

---

> > ### Comment · Reviewer_rWBe · 2023-08-11
> > **reply to authors**
> >
> > This is to acknowledge that I have read the author's response to my review. Thanks

---

### Official Review · Reviewer_QP8B · 2023-07-07

**Soundness:** 3 good
**Presentation:** 2 fair
**Contribution:** 2 fair
**Rating:** 4
**Confidence:** 3

**Summary:**

The authors develop specific heat diffusion kernels for manifold embedding.
They define a heat kernel based on the Varadha formula.
The authors try to weave some connections with existing methods of DR such as PHATE, t-SNE, etc.
They run some experiments

**Strengths:**

The paper is well written except for the organization, clarity, and self-containedness.
The presented developments about the geodesic distance and the heat diffusion kernel are of interest for the manifold learning community.
The authors quite honestly acknowledge some of the limitations of their work.

**Weaknesses:**

The writing style is too dense, poorly structured, with a succession of (too) short sections, referring to each other too many times, and often missing key information.
Isomap is not cited and not compared to (in Fig. 1 for instance; later it seems that Isomap is actually used, without saying the name and still without a bibliograpcal reference, or is 'shortest path' with stress-based MDS instead of classical MDS? This can be referenced too, most probably).
The mathematical notation is misleading and unusual for manifold learning (x and y in the same space! use x_i and x_j ?).
The experiments have unclear/contradictory objectives: to detect manifold structures and clusters? The COIL-20 or 100 data set should then be considered. The results in Fig. 1 are not convincing at all: Isomap does better. In experiments related to clustering, the reader might always question the purposefulness of assessing cluster separation in methods which are local and depending on graphs; a disconnected graph somehow automatically guarantees cluster separation. On the other hand, the arrangement of clusters with respect to each other might actually be more relevant. The clustering experiments on Swiss roll and tree are far-fetched (use COIL-20?), with completely artificial clusters. Over-expression of clusters or even spurious clusters have been reported with many 'local' methods of manifold learning. Assessing the preservation of distances or neighbourhoods might be less of a stretch between the focus on geodesic distances and the current interest for clustering with those methods in communities of practitioners.


**Questions:**

What is V(x,sqrt(t)) ? please define and/or give an intuitive interpretation. The section about Varadhan's formula seems to end prematurely. Probably Vradhan could gain to be detailed more while the subsequent paragraphs in Preliminaries could be shortened or dropped (except for the quick reminder about stress-based MDS).
Section 3 about related work is going too fast and lacks details (reminder of P for PHATE? drop SNE that is quite well known).
In section 4, if distance order preserving matters, why not compare to non-metric MDS too?
Can you clarify Section 4? Definition 4.1 is cryptic and unintuitive; what is the meaning of sigma?
In proposition 4.1, what is |x-y| ? absolute value? distance like ||.||_2 elsewhere?
Why come back on MDS? Merge the presentation of MDS and its specific weighting?
Section 5 is also wanting to embrace everything within limited space: alonger version in appendix and more details about the core contribution of the paper
Ref 33: Cédric Villani has a twin brother?

**Limitations:**

Yes. They have. No potential negative societal impact here.

---

> ### Author Rebuttal · Authors · 2023-08-10
>
> We would like to thank the reviewers for their constructive comments on our manuscript. Below we will address the main concerns of the reviewer.
>
> >The writing style is too dense, poorly structured, with a succession of (too) short sections, referring to each other too many times, and often missing key information.
>
> Thanks for this observation, we will try to improve the connection between paragraphs to make reading smoother.  See also our comments in the section **On the reviewer’s questions** for more detailed modifications on the organization of the manuscript.
>
> **About Isomap**
>
> In our submission, in particular in Tab.1, we referred to Isomap as the shortest-path method since we only used the distance matrix (i.e. only the shortest path). Indeed, Isomap is MDS done on shortest path distances between points on an affinity graph. However, we improved the clarity of the paper by linking to the shortest path distance to Isomap, and referencing Isomap explicitly. We emphasize that while Isomap indeed finds manifold distances in noiseless settings, the performance degrades in noisy data where the data graph is not reliable. For instance, in the embryoid body data, Isomap muddles many of the differentiation structures as also noted in the PHATE paper [Moon et al. 2019] (see Fig.1 in the global response's PDF). The same observation is made in our new quantitative results on Isomap (see tables in the global response's PDF)
>
> **About misleading notation**
>
> >The mathematical notation is misleading and unusual for manifold learning (x and y in the same space! use x_i and x_j ?).
>
> We have uniformized the notation, we use $x_i,x_j$ for points in the ambient space and $y_i,y_j$ for points in the embedded space.
>
> **About the apparent trade off between manifold structure learning and clusters.**
>
> As discussed in our general response, we want to emphasize that the distance preservation is not at odds with the cluster structure preservation. For clusters defined by the geodesic on a manifold, methods like HeatGeo, which are aimed at leveraging the underlying geometry of the data, should do better. In particular, as the reviewer points out, our method would remain faithful to the relationships between different clusters, in contrast to methods like t-SNE or UMAP, that may distort or inflate the separation. This is further illustrated on EMD interpolation experiment (Table 3) that aims at evaluating the quality of the relationship between different regions of the data. Experiments like the Gaussians on the swiss roll is a prominent example of clustering on a continuous manifold. In this case, the distance preservation and the cluster structure preservation are concordant objectives.
>
> That said, some clustering tasks may indeed not perfectly fit the underlying manifold assumption, such as MNIST or COIL. We appreciate the reviewer’s suggestion of adding another high dimensional dataset, to put our method in perspective and we have added comparisons on MNIST and COIL20. You can see the full table in the PDF attached to the global response, this table will also be added to the main body of the text. On these datasets, t-SNE shows higher homogeneity and adjusted mutual information.
>
> **On the reviewer’s questions**
>
> >What is $V(x,sqrt(t))$ ?
>
> It is the volume of a ball of radius $\sqrt{t}$ centered at $x$, we added the definition.
>
> > Probably Vradhan could gain to be detailed more …
>
> In the appendix C.3, we give a few examples on the application of the two-sided inequality and Varadhan’s formula, this would probably help the reader to understand. However we did not reference that section in the main body of the text, we have now fixed this.
>
> > the subsequent paragraphs in Preliminaries could be shortened or dropped…
>
> We appreciate the reviewer’s advice on restructuring the preliminaries section and the related work. We believe we can shorten the subsection “Fast computation of Heat diffusion” and move some details to the appendix, and use the additional space to improve the fluidity of the related work section.
>
> > In section 4, if distance order preserving matters, why not compare to non-metric MDS too?
>
> We added the comparison to non-metric MDS. The results are included in the PDF attached to the global response. Overall our method does better.
>
> > Can you clarify Section 4?...
>
> We are working on improving the clarity of section 4. In particular, we added more intuition behind definition 4.1, and how it relates to the two-sided heat kernel inequality. We also fixed the notation on the norm $|\cdot|$ vs $\||\cdot\||_2$.
>
> >Section 5 is also wanting to embrace everything within limited space…
>
> Section 5 is a bit dense because it had to fit within the page limit. However in the appendix we provide the full details of the proof (Sec. A.2) and the unifying framework (Sec. A.1). In the appendix, before each proposition, we are adding a small paragraph explaining the results.
>
> >Ref 33: Cédric Villani has a twin brother?
>
> Thank you for pointing this out, we fixed the reference.
>
> We would like to thank the reviewer for their review of our paper. We believe we have answered all the great points raised by the reviewer in our author response. We respectfully ask the reviewer to reconsider their impression of the paper and potentially improve the given score if the raised concerns have been allayed. We thank the reviewer again for their time and we are also happy to answer any further questions that arise!

---

> ### Author Response · Authors · 2023-08-15
> **Did we address all the concerns raised ?**
>
> Dear Reviewer QP8B,
>
> Thank you for reviewing our paper. May we kindly ask if our answers addressed all your concerns ? Please let us know if additional clarifications are needed.
>
> Thank you so much,
>
> The authors

---

> > ### Comment · Reviewer_QP8B · 2023-08-18
> >
> > Thanks for the improvements and answers.

---

### Official Review · Reviewer_B5m7 · 2023-07-08

**Soundness:** 3 good
**Presentation:** 3 good
**Contribution:** 2 fair
**Rating:** 6
**Confidence:** 4

**Summary:**

The paper introduces a dimensionality reduction technique based on heat kernels and called HeatGeo. The main idea is to learn a mapping that projects points to a low-dimensional space (ideally 2 or 3-dimensional for visualization) so that relative distances between pairs of points satisfy the same order as distances on some (complete Riemannian) manifold that uses some distance based on the heat kernel.

**Strengths:**

Originality: The idea of using heat kernels for dimensionality reduction is not new (e.g. Belkin & Niyogi [1] that proposed Laplacian eigenmaps). This paper proposes a generalization of previous techniques by exploiting Varadhan's formula (Eq. (1)) and by bounding the calculated distance on a low-dimensional complete Riemannian manifold. They show that the ordering of distances in the high-dimensional space tends to be preserved on the low-dimensional manifold.

Quality and Clarity: The paper is well-written and mentions different approximation techniques that can be used to approximate the computation of the kernel matrix since the algorithmic complexity is cubic in the number of points. Overall, the paper shows that the proposed approach tends to preserve clusters and the ordering of distances in the low-dimensional space better than standard approaches such as t-SNE and UMAP.

Significance: The significance of the approach is difficult to quantify. As most dimensionality reduction techniques that are often used for qualitative evaluation in the machine learning literature, the proposed approach can improve the understanding of a data set with a visualization that preserves the original general structure.

**Weaknesses:**

I do not see a particular weakness in the formulation of the approach which has a nice theoretical justification.

The main weakness of the paper is that it mainly focuses on dimensionality reduction and visualization which already have widely used techniques (e.g. t-SNE and UMAP) that are used only for qualitative evaluation in practice.

**Questions:**

I have no questions.

---

> ### Author Rebuttal · Authors · 2023-08-09
>
> We thank the reviewer for their encouraging review. Below we respond to the main weakness raised by the reviewer.
>
> >The main weakness of the paper is that it mainly focuses on dimensionality reduction and visualization which already have widely used techniques (e.g. t-SNE and UMAP) that are used only for qualitative evaluation in practice.
>
>
> We agree with the reviewer that several methods have been proposed for data representation learning. A major motivation for our work is to unify these different methods through the lens of differential geometry (with Varadhan formula and Harnack inequality). We believe this helps understand why and when these methods work and how their algorithmic difference can impact the resulting embeddings. For instance, neither t-SNE nor UMAP preserve distances, and this has an impact on the visualization as opposed to methods that use diffusion or other discretizations of heat flow on the manifold which is explicitly connected to heat diffusion.
>
> While these methods can be assessed qualitatively, by inspecting the low dimensional embeddings, we aimed at also providing a rigorous quantitative evaluation. We did so by comparing the estimated distance matrices against ground truth, and by assessing whether the embeddings conserved the underlying structure of the data (through cluster evaluation and EMD interpolation). These results showed that our method, supported by its strong theoretical foundations, along with closely related approaches (e.g. PHATE), produced embeddings that were more faithful to the inherent structure of the data.

---

> > ### Comment · Reviewer_B5m7 · 2023-08-14
> > **I have read the authors' rebuttal.**
> >
> > I have read the authors' rebuttal.

---

### Official Review · Reviewer_7Dv3 · 2023-07-14

**Soundness:** 3 good
**Presentation:** 2 fair
**Contribution:** 3 good
**Rating:** 5
**Confidence:** 3

**Summary:**

This article deals with representation learning and dimensionality reduction of high dimensional noisy datasets. The proposed method relies on heat diffusion. More precisely, assuming their data can be viewed as Riemannian manifolds and based on Varadhan’s formula, the authors define the notion of heat-geodesic dissimilarity. According to the authors, using the heat equation guarantees the conservation of geodetic distances between the raw data and its integrated counterpart. This dissimilarity is then used through multidimensional scaling (MDS) to define their so-called heat geodesic embedding.

Afterward, the authors theoretically compare their method to the start-of-the-are ones. In particular, in the appendix, they attempt to create a generalized framework for dimensionality reduction methods.

Last, they conduct numerical experiments to test their algorithm on different data types. In particular, they want to ensure that their method effectively preserves geodetic distances. They also seek to ensure that the inherent data structures are conserved: Preserving clusters or data temporality, for instance.

**Strengths:**

- Interesting idea, based on the solid formalist of Riemannian manifolds.
- Efforts to unify different embedding approaches (especially in the appendix) with comparisons with existing methods.
- Numerical experiments demonstrating the benefits of heat geodesic embedding for understanding data.


**Weaknesses:**

There are typos and undefined notations in several places (see below). This is a disservice to the paper.


**Questions:**

Some comments:
- Line 66: $n$ not defined.
- Line 67: I think it should be $\frac{n}2$ instead of $2n$.
- Eq (2): $V(x,\sqrt{t})$ not defined.
- Paragraph starting line 79 (Graph construction and diffusion): $n$ is not defined, although we understand it is the sample size in context.
- Line 88: Notation $[n]$ not defined.
- Line 94: Add references when referring to literature.
- Line 160: Should the $n$ not be a $d$ here?
- Line 181: they use the a distance... -> the _or_ a.
- Line 304: Strange line break.

**Limitations:**

Not concerned.

---

> ### Author Rebuttal · Authors · 2023-08-09
>
> We would like to thank the reviewer for their suggestions, and we agree that the quality of the writing could be improved. We did a thorough pass on the manuscript, and corrected some typos including the ones mentioned. We also added the missing definitions. We believe that these minor corrections help improve the overall clarity of the manuscript, and respectfully ask the reviewer to reconsider their impression of the paper and potentially improve the given score if the raised concerns have been allayed. We thank the reviewer again for their time and we are also happy to answer any further questions that arise!

---

### Author Rebuttal · Authors · 2023-08-10

We would like to thank the reviewers for their time and valuable feedback when reviewing our paper. These suggestions led us to make a few modifications, which we believe greatly improved the quality of the manuscript. We now respond below to the main clarification points grouped by theme.

- **On our key contributions in unification of embeddings based on the idea of heat diffusion** We wish to highlight that one of our key contributions is to examine and interpret manifold-preserving embeddings based on properties of heat diffusion on the manifold. The key results we use to do this from geometry are Varadhan’s formula and Harnack’s inequality (which gives a two-sided bound on manifold distances based on heat diffusion). In particular, this perspective provides a better understanding of PHATE, diffusion maps, and tSNE. Then, we propose an embedding based on the heat kernel that makes this link more direct, i.e. the methods PHATE, diffusion maps and tSNE all use a random walk operator that indirectly models heat diffusion in discrete time whereas the heat kernel is a more direct continuous embodiment. As noted by reviewer Yhnd, this indeed makes our embedding “somewhat similar” (on purpose) to other manifold-preserving embeddings as it is a unification of the theory in those methods.


- **On distance preservation and clusters**
The reviewers Yhnd and QP8B were skeptical about the clustering evaluation. In this manuscript we are proposing and unifying manifold-distance preserving embeddings. We want to note that distance preservation is in no way at odds with cluster structure preservation. Indeed, clusters are formed by data points that are close (low manifold distances) and separations between clusters are indicated by high distance. In fact, the distance preserving methods also keep the relationships as our method correlates well with the intrinsic distance on the manifold (see Tab.1). Indeed, a method that would perfectly recover intrinsic distances should also recover the resulting clusters perfectly. As such, for clusters defined by the geodesic distance on a manifold, methods like HeatGeo should do better, and indeed PHATE [2] showed superior clustering performance as compared to other methods like UMAP/tSNE which do not preserve distances. This is further illustrated in our EMD interpolation experiment (Table 3) that aims at evaluating the quality of the relationship between different regions of the data, and in Table 1 that reports the correlation between the different distances and the geodesic.
In our clustering experiments, we test a single cell PBMC dataset (known to have clusters), a mixture of Gaussian distributions on the swiss roll, and a tree with five branches. We have added experiments on MNIST and COIL-20; two datasets for which the manifold hypothesis may not be respected. Our methods show greater accuracy on the swiss roll and tree datasets, and competitive performance on the other datasets (see the full results in Tab.1 of the attached PDF).

- **Additional comparisons to ISOMAP and non-metric MDS.** The reviewers Yhnd and QP8B mentioned the lack of comparison with Isomap, and reviewer QP8B also suggested using non-metric MDS. We followed their advice and added quantitative comparisons with non-metric MDS, and Isomap. For the MDS step in Isomap, we used the same optimization algorithm as our method, in order to accurately compare the two algorithms. Overall our method HeatGeo performs better than non-metric MDS and Isomap (see the preliminary results in the attached PDF). We will add visualizations for these methods in the appendix. This is to be expected in noisy data since shortest path distances (which Isomap uses) are highly sensitive to noise, and non-metric MDS ordinally preserves Euclidean and not manifold distances.

- **On computational complexity.**
Reviewer Yhnd asked for a discussion on complexity. We do mention the complexity of the approximation of the heat kernel (line 114), and we will add a discussion on the total complexity in the manuscript. While the most costly step of our current implementation constructs a full distance matrix for quadratic $O(n^2)$ cost, this is not strictly necessary. Similar to other methods, if we are willing to trade off accuracy for speed, we can truncate at some radius around each point to get a sparse distance matrix. If we do so then our algorithm will be $O(E)$ where E is the number of distances smaller than the chosen radius. This would give the same complexity (both time and space) as UMAP and TSNE algorithms, a log-linear-time sparse graph construction, a log-linear Chebyshev diffusion, followed by a (difficult to analyze) linear SGD optimization. We note that we use the stochastic gradient descent metric-MDS [1], which is much faster than using e.g. the SMACOF algorithm. In general our methods can also be made faster using landmarking [2, 3]. We also ran additional experiments to compare the computation time of the different methods. The preliminary results can be found in Tab. 3 of the attached pdf, which shows that our method is the fastest when using Chebyshev polynomials. We will show additional results by varying the number of datapoints.

- **We improved the readability and structure of the manuscript.** We did a thorough revision on the manuscript and corrected typos or misprints, improving the overall clarity of the manuscript.

[1] Zheng, J. X., Pawar, S., & Goodman, D. F. (2018). Graph drawing by stochastic gradient descent. IEEE transactions on visualization and computer graphics, 25(9), 2738-2748.

[2] Moon, K. R., van Dijk, D., Wang, Z., Gigante, S., Burkhardt, D. B., Chen, W. S., ... & Krishnaswamy, S. (2019). Visualizing structure and transitions in high-dimensional biological data. Nature biotechnology, 37(12), 1482-1492.

[3] Silva, V., & Tenenbaum, J. (2002). Global versus local methods in nonlinear dimensionality reduction. Advances in neural information processing systems, 15.

---

> ### Comment · Reviewer_Yhnd · 2023-08-14
> **MNIST**
>
> > We have added experiments on MNIST and COIL-20; two datasets for which the manifold hypothesis may not be respected. Our methods show greater accuracy on the swiss roll and tree datasets, and competitive performance on the other datasets (see the full results in Tab.1 of the attached PDF).
>
> Thanks for doing this experiment. In this new Table one can clearly see that on both MNIST and COIL, t-SNE wins with a large margin. I would not call it "competitive performance" :) Just to be clear: I don't think it hurts your paper in any way! Your paper is clearly about manifold learning and MNIST is not a manifold. I would just like that this is clearly described, and that the method is not oversold.

---

### Decision · Program_Chairs · 2023-09-21

**Decision:**

Accept (poster)

**Comment:**

Reviews on this paper are mixed but tending positive; the most positive and most negative reviews both have some aspects that make them less than reliable.

In the end, the AC decided to suggest accepting this paper.  Although the approach here is clearly related to many others in the diffusion maps literature, the idea certainly caught the interest of all the reviewers and generated discussion.  Moreover, the mathematical formulation here is different than motivation for past diffusion maps constructions of embeddings.